# Functional synergy between the Munc13 C-terminal C$_1$ and C$_2$ domains

Xiaoxia Liu[1,2,3†], Alpay Burak Seven[1,2,3†], Marcial Camacho[4], Victoria Esser[1,2,3], Junjie Xu[1,2,3], Thorsten Trimbuch[4], Bradley Quade[1,2,3], Lijing Su[1,2,3], Cong Ma[5,6], Christian Rosenmund[4], Josep Rizo[1,2,3*]

[1]Department of Biophysics, University of Texas Southwestern Medical Center, Dallas, United States; [2]Department of Biochemistry, University of Texas Southwestern Medical Center, Dallas, United States; [3]Department of Pharmacology, University of Texas Southwestern Medical Center, Dallas, United States; [4]Department of Neurophysiology, NeuroCure Cluster of Excellence, Charité-Universitätsmedizin Berlin, Berlin, Germany; [5]Key Laboratory of Molecular Biophysics of the Ministry of Education, Huazhong University of Science and Technology, Wuhan, China; [6]College of Life Science and Technology, Huazhong University of Science and Technology, Wuhan, China

**Abstract** Neurotransmitter release requires SNARE complexes to bring membranes together, NSF-SNAPs to recycle the SNAREs, Munc18-1 and Munc13s to orchestrate SNARE complex assembly, and Synaptotagmin-1 to trigger fast Ca$^{2+}$-dependent membrane fusion. However, it is unclear whether Munc13s function upstream and/or downstream of SNARE complex assembly, and how the actions of their multiple domains are integrated. Reconstitution, liposome-clustering and electrophysiological experiments now reveal a functional synergy between the C$_1$, C$_2$B and C$_2$C domains of Munc13-1, indicating that these domains help bridging the vesicle and plasma membranes to facilitate stimulation of SNARE complex assembly by the Munc13-1 MUN domain. Our reconstitution data also suggest that Munc18-1, Munc13-1, NSF, αSNAP and the SNAREs are critical to form a 'primed' state that does not fuse but is ready for fast fusion upon Ca$^{2+}$ influx. Overall, our results support a model whereby the multiple domains of Munc13s cooperate to coordinate synaptic vesicle docking, priming and fusion.

*For correspondence: Jose.Rizo-Rey@UTSouthwestern.edu

†These authors contributed equally to this work

## Introduction

The release of neurotransmitters by Ca$^{2+}$-evoked synaptic vesicle exocytosis is a key event for communication between neurons and involves several steps, including vesicle docking at presynaptic active zones, a priming reaction(s) that leaves the vesicles ready for release, and fast Ca$^{2+}$-triggered fusion of the vesicle and plasma membranes (*Sudhof, 2013*). Studies of the complex machinery that controls release have shown that eight proteins are particularly important and have established defined roles for them (*Rizo and Xu, 2015*; *Jahn and Fasshauer, 2012*; *Brunger et al., 2015*; *Sudhof and Rothman, 2009*): i) the soluble N-ethylmaleimide-sensitive factor attachment protein receptors (SNAREs) syntaxin-1, SNAP-25 and synaptobrevin bring the membranes together by forming a four-helix bundle called SNARE complex (*Sollner et al., 1993*; *Poirier et al., 1998*; *Sutton et al., 1998*), which is critical for membrane fusion (*Hanson et al., 1997*); ii) N--ethylmaleimide sensitive factor (NSF) and soluble NSF attachment proteins (SNAPs; no relation to SNAP-25) disassemble the SNARE complex (*Sollner et al., 1993*) to recycle the SNAREs (*Mayer et al., 1996*; *Banerjee et al., 1996*); iii) Munc18-1 and Munc13s orchestrate SNARE complex

**eLife digest** In the brain, neurons communicate with each other using small molecules called neurotransmitters. Electrical signals in one neuron trigger the release of the neurotransmitters, which then bind to receptor proteins on another neuron nearby. Neurotransmitters are packaged into small compartments called synaptic vesicles and are released from the neuron when these vesicles fuse with the membrane that surrounds the cell. Many proteins are involved in regulating this process to ensure that neurotransmitters are released at the right place and time.

A large protein called Munc13 plays an important role in the release of neurotransmitters. It contains many different regions, including a long domain called MUN and three additional domains called $C_1$, $C_2B$ and $C_2C$ among others. However, it is not clear how all these domains work together to control neurotransmitter release. Here Liu, Seven et al. address this question using purified proteins inserted into membranes as well as experiments in neurons from mice. The experiments show that the $C_1$, $C_2B$ and $C_2C$ domains all play key roles in neurotransmitter release. Together with the MUN domain, these three domains help to form bridges between synaptic vesicles and the membrane surrounding the neuron. These bridges could help other proteins involved in neurotransmitter release to form a group that induces vesicle fusion.

Liu, Seven et al.'s findings also suggest that Munc13 proteins cooperate with other proteins to form a 'primed' state in which a synaptic vesicle is ready to rapidly fuse with a neuron's membrane when triggered to do so by an electrical signal. A future challenge is to find out how the proteins that form this primed state promote vesicle fusion.

assembly, which involves initial binding of Munc18-1 to a self-inhibited 'closed' conformation of syntaxin-1 (*Dulubova et al., 1999*; *Misura et al., 2000*) and opening of syntaxin-1 by Munc13 (*Richmond et al., 2001*; *Ma et al., 2011*; *Yang et al., 2015*); iv) and Synaptotagmin-1 (Syt1) acts as the $Ca^{2+}$ sensor that triggers fast release (*Fernandez-Chacon et al., 2001*), likely via interactions with both membranes (*Arac et al., 2006*) and with the SNARE complex (*Brewer et al., 2015*; *Zhou et al., 2015*).

Reconstitution experiments (*Weber et al., 1998*) have contributed to establishing some of these key concepts, providing a powerful tool to study the mechanism of synaptic vesicle fusion (*Brunger et al., 2015*). It is now clear that synaptobrevin-liposomes can fuse with syntaxin-1-SNAP-25 liposomes under some conditions but not others, and that fusion is stimulated to different degrees by Syt1, Munc18-1 or Munc13-4 (*Weber et al., 1998*; *van den Bogaart et al., 2010*; *Tucker et al., 2004*; *Yu et al., 2013*; *Kyoung et al., 2011*; *Lee et al., 2010*; *Boswell et al., 2012*; *Parisotto et al., 2012*), but such fusion is abolished by NSF and αSNAP because they disassemble syntaxin-1-SNAP-25 complexes (*Weber et al., 2000*; *Ma et al., 2013*). However, inclusion of Munc18-1 and a Munc13-1 fragment enable fusion at least in part because they protect against the disassembly activity of NSF-αSNAP while coordinating SNARE complex assembly (*Ma et al., 2013*). These results explained the essential nature of Munc18-1 and Munc13s for neurotransmitter release (*Verhage et al., 2000*; *Richmond et al., 1999*; *Varoqueaux et al., 2002*; *Aravamudan et al., 1999*) and correlated with earlier studies of the role of the HOPS tethering complex in yeast vacuolar fusion (*Xu et al., 2010*).

Despite these advances, fundamental questions remain about the mechanism of neurotransmitter release, in particular regarding the functions of Munc13s (*Rizo and Xu, 2015*). These large proteins of presynaptic active zones contain a variable N-terminal region that in some isoforms include a $C_2$ domain (the $C_2A$ domain), and a highly conserved C-terminal region that includes (see *Figure 1A* for Munc13-1): i) a $C_1$ domain involved in diacyglycerol (DAG)-phorbol ester-dependent augmentation of release (*Rhee et al., 2002*; *Basu et al., 2007*); a $C_2B$ domain that regulates release probability and modifies short term plasticity through its $Ca^{2+}$- and phosphatidylinositolphosphate-binding activities (*Shin et al., 2010*); a MUN domain that is key for the crucial function of Munc13 in release (*Basu et al., 2005*) and mediates the activity of Munc13 in opening syntaxin-1 (*Richmond et al., 2001*; *Ma et al., 2011*; *Yang et al., 2015*); and a $C_2C$ domain that is also important for release (*Madison et al., 2005*; *Stevens et al., 2005*) and is not predicted to bind $Ca^{2+}$ but may bind

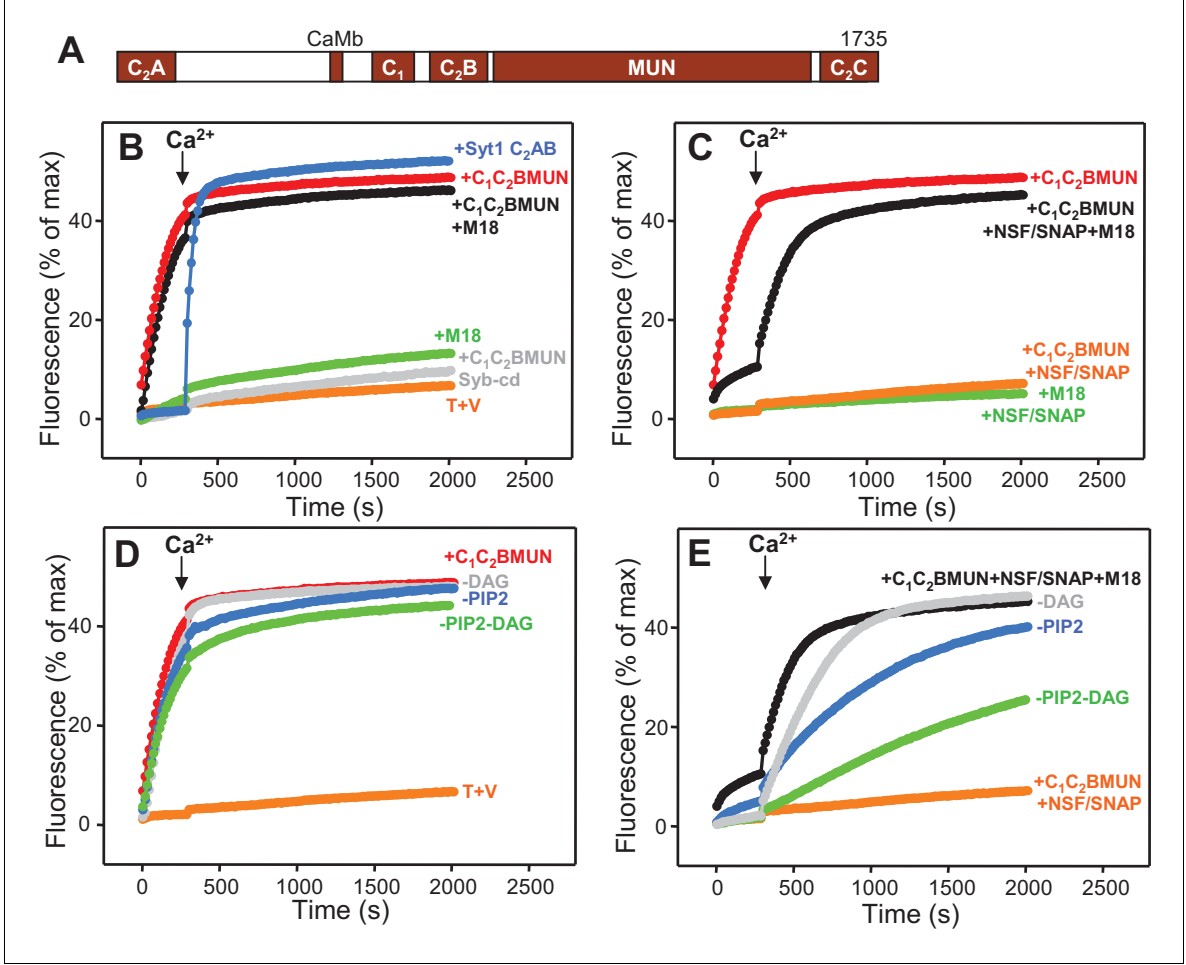

**Figure 1.** Munc13-1 $C_1C_2BMUN$ strongly stimulates lipid mixing between V- and T-liposomes. (**A**) Domain diagram of Munc13-1. CaMb = calmodulin-binding sequence. (**B–C**) Lipid mixing assays between V- and T-liposomes alone (T+V) or in the presence of different combinations of Munc13-1 $C_1C_2BMUN$, Syt1 $C_2AB$ fragment, Munc18-1 (M18), NSF-$\alpha$SNAP (NSF/SNAP) and synaptobrevin cytoplasmic domain (Syb-cd). T-liposomes contained 1% DAG and 1% PIP$_2$. (**D–E**) Analogous lipid mixing assays performed in the presence of $C_1C_2BMUN$ (**D**) or $C_1C_2BMUN$ plus Munc18-1 and NSF-$\alpha$SNAP (NSF/SNAP) (**E**) with T-liposomes containing 1% DAG and 1% PIP$_2$, 1% PIP$_2$ (-DAG), 1% DAG (-PIP2) or no DAG and PIP$_2$ (-DAG-PIP2). Controls of T+V (**D**) or T+V in the presence of $C_1C_2BMUN$ plus NSF-$\alpha$SNAP (NSF/SNAP) (**E**), both with T-liposomes containing 1% DAG and 1% PIP$_2$, are shown in orange. All experiments were started in the presence of 100 $\mu$M EGTA, and Ca$^{2+}$ (600 $\mu$M) was added after 300 s.

The following figure supplement is available for figure 1:

**Figure supplement 1.** Quantification of the lipid mixing experiments of *Figure 1*.

phospholipids because this is a common property of C$_2$ domains (*Rizo and Sudhof, 1998*). The central function of Munc13s in neurotransmitter release was initially associated to an essential role in vesicle priming (*Augustin et al., 1999*), but later studies that used stringent definitions of vesicle docking (see discussion) uncovered a critical role for Munc13s in docking that was attributed to their activity in mediating SNARE complex assembly (*Weimer et al., 2006*; *Hammarlund et al., 2007*; *Imig et al., 2014*). However, it is unknown whether Munc13s participate in upstream interactions that might help bridging the two membranes to promote docking and priming. This possibility is attractive because the MUN domain is related to tethering factors involved in diverse forms of membrane traffic (*Pei et al., 2009*; *Li et al., 2011*) and is flanked by domains with demonstrated or potential lipid-binding properties. Moreover, while there is evidence that Munc13s modulate release probability and have a role beyond docking [e.g. (*Rhee et al., 2002*; *Hammarlund et al., 2007*;

*Shin et al., 2010*)], it is unclear whether they form part of the primed complex after SNARE complex assembly, influencing membrane fusion downstream of priming.

To shed light into these questions, we have used a combination of reconstitution and dynamic light scattering (DLS) assays together with electrophysiological experiments. Our results show that both the $C_1$-$C_2B$ region and the $C_2C$ domain of Munc13-1 play important functions in release, and suggest that these domains help bridging synaptic vesicles to the plasma membrane, facilitating the activity of the Munc13-1 MUN domain in promoting SNARE complex assembly. Moreover, our results indicate that the neuronal SNAREs, Munc18-1, NSF, $\alpha$SNAP and a Munc13-1 fragment including the $C_1$, $C_2B$, MUN and $C_2C$ domains ($C_1C_2BMUNC_2C$) are sufficient to generate a 'primed' state that is ready to trigger fast membrane fusion upon addition of $Ca^{2+}$, thus resembling the primed state of synaptic vesicles.

## Results

### Munc13-1 $C_1C_2B$MUN strongly stimulates SNARE-dependent lipid mixing

In experiments that followed our recent reconstitution study (*Ma et al., 2013*) and were directed at analyzing how different factors affect the efficiency of membrane fusion, we first analyzed lipid mixing between synaptobrevin-liposomes (V-liposomes) and syntaxin-1-SNAP-25 liposomes (T-liposomes) by monitoring de-quenching of the fluorescence of NBD-labeled lipids incorporated in the synaptobrevin-liposomes (*Weber et al., 1998*) (*Figure 1*). These experiments were initiated in the absence of $Ca^{2+}$, and $Ca^{2+}$ was added at 300 s to examine the $Ca^{2+}$-dependence of the results. The liposomes contained a synaptic-like lipid composition and, unless otherwise specified, the T-liposomes included DAG and $PIP_2$, which activate the Munc13-1 $C_1$ and $C_2B$ domains (*Ma et al., 2013*). We illustrate the reproducibility of some of the data in Supplementary Figures, showing the quantification of the data at 500 s.

In the NBD de-quenching assays we observed that lipid mixing between V- and T-liposomes is strongly stimulated by a Munc13-1 fragment spanning its $C_1$, $C_2B$ and MUN domains ($C_1C_2B$MUN) (*Figure 1B*; red data), which contrasts with the smaller stimulation observed earlier [Figure S8 of (*Ma et al., 2013*)]. We note that we have been able to reproduce all other results from this previous study and we speculate that the sample of $C_1C_2B$MUN used for the experiments of Figure S8 of *Ma et al. (2013)*, which were performed at the end of that study, might have been partially inactivated during storage in the freezer, as we have reproduced the results shown in *Figure 1B* in more than 20 subsequent reconstitution experiments employing at least five different preparations of $C_1C_2B$MUN. The enhancement of lipid mixing induced by $C_1C_2B$MUN was independent of $Ca^{2+}$ and was SNARE-dependent, as it was strongly impaired by addition of the cytoplasmic region of synaptobrevin (Syb-cd) (*Figure 1B* and *Figure 1—figure supplement 1A*). The extent of lipid mixing between V- and T-liposomes in the presence of $C_1C_2B$MUN was comparable to that caused by a soluble fragment spanning the two $C_2$ domains of Syt1 ($C_2AB$ fragment) in the presence of $Ca^{2+}$; in contrast, Munc18-1 had only a small stimulatory effect on lipid mixing, and did not enhance the stimulation caused by $C_1C_2B$MUN (*Figure 1B* and *Figure 1—figure supplement 1A*).

As expected, addition of NSF-$\alpha$SNAP abolished the strong stimulatory effect of $C_1C_2B$MUN but lipid mixing was highly efficient again when both $C_1C_2B$MUN and Munc18-1 were added in the presence of NSF-$\alpha$SNAP (*Figure 1C* and *Figure 1—figure supplement 1B*), consistent with the notion that $C_1C_2B$MUN and Munc18-1 mediate SNARE complex assembly in an NSF-$\alpha$SNAP-resistant manner (*Ma et al., 2013*). Note that these experiments did not include Syt1 $C_2AB$ and yet lipid mixing was $Ca^{2+}$-dependent in the presence of $C_1C_2B$MUN, Munc18-1 and NSF-$\alpha$SNAP. Moreover, removal of DAG, $PIP_2$ or both from the T-liposomes caused increasingly stronger impairments of lipid mixing in these experiments but had much milder effects on the stimulation of lipid mixing caused by $C_1C_2B$MUN in the absence of NSF-$\alpha$SNAP (*Figure 1D,E* and *Figure 1—figure supplement 1C,D*). These data show that the effect of Munc13-1 $C_1C_2B$MUN alone on lipid mixing arises from a property that is largely independent of $Ca^{2+}$, DAG and $PIP_2$, whereas the lipid mixing observed in the more complete reconstitutions including $C_1C_2B$MUN, Munc18-1 and NSF-$\alpha$SNAP is stimulated by

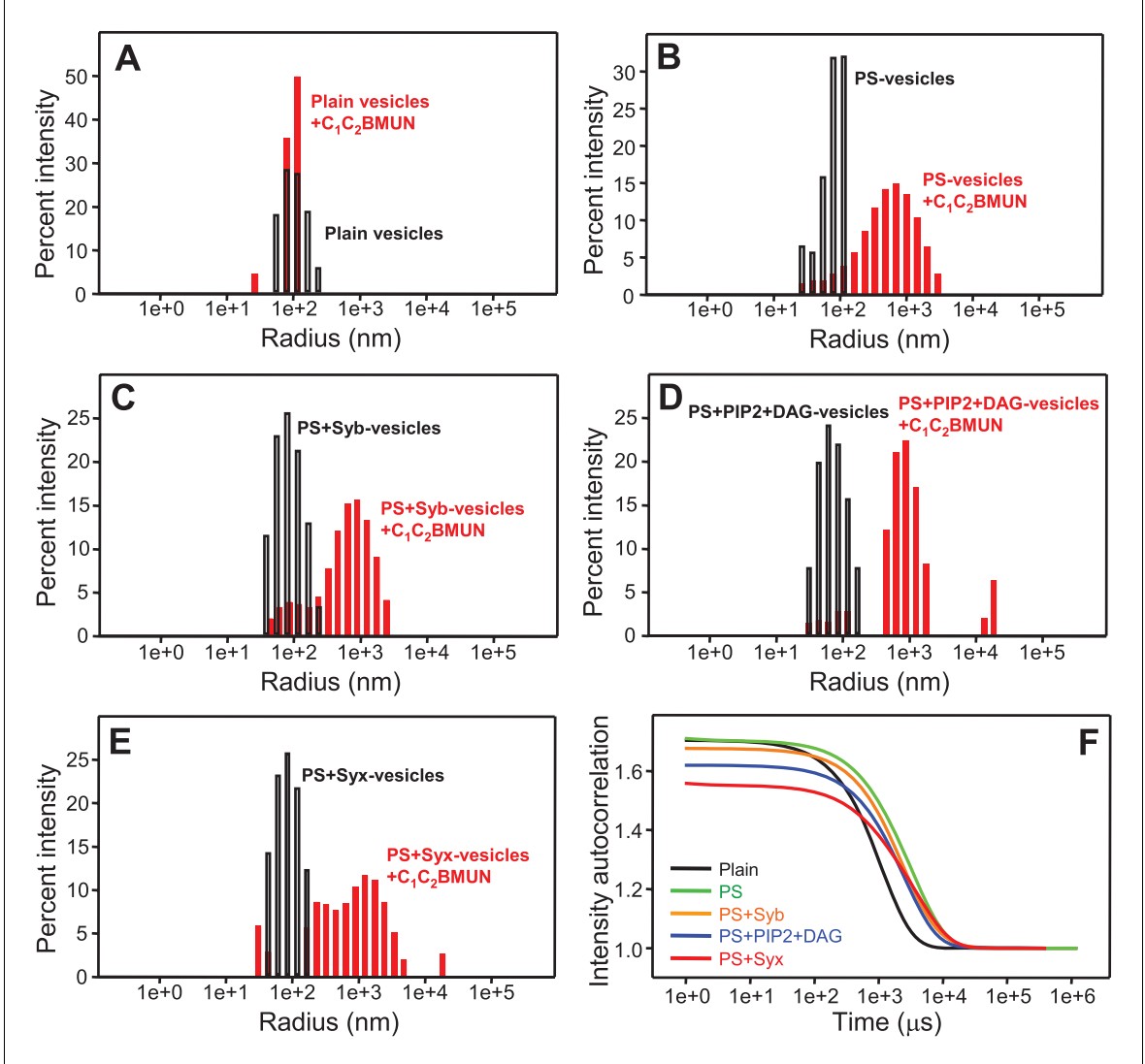

**Figure 2.** Munc13-1 $C_1C_2BMUN$ clusters PS-containing liposomes. (**A–E**) The particle size in samples containing phospholipid vesicles alone (gray bars) or after incubation with Munc13-1 $C_1C_2BMUN$ for 5 min (red bars) in the absence of $Ca^{2+}$ was measured by DLS. The liposomes had a standard lipid composition including no PS (**A**), PS (**B**), PS and synaptobrevin (**C**), PS+DAG+PIP$_2$ (**D**) or PS and syntaxin-1 (**E**). (**F**) Intensity autocorrelation curves corresponding to the experiments shown in (**A–E**) after incubation with Munc13-1 $C_1C_2BMUN$ for 5 min.

The following figure supplement is available for figure 2:

**Figure supplement 1.** $Ca^{2+}$ does not stimulate liposome clustering by $C_1C_2BMUN$ strongly.

$Ca^{2+}$, DAG and PIP$_2$, thus exhibiting properties that are more similar to those of neurotransmitter release.

## Munc13-1 $C_1C_2BMUN$ clusters phosphatidylserine-containing liposomes

The ability of Syt1 $C_2AB$ to bind simultaneously to two membranes in a $Ca^{2+}$-dependent manner (*Arac et al., 2006*) underlies at least in part its activity in stimulating SNARE-dependent lipid mixing (*Tucker et al., 2004*; *Xue et al., 2008*). Hence, we tested whether Munc13-1 $C_1C_2BMUN$ is also able to bridge two membranes by monitoring the formation of liposome clusters by DLS. While $C_1C_2BMUN$ did not cluster plain liposomes lacking phosphatidylserine (PS), dramatic increases in particle size observed by DLS revealed efficient clustering of PS-containing vesicles caused by $C_1C_2BMUN$ (*Figure 2A,B*). Inclusion of synaptobrevin, DAG+PIP$_2$ or syntaxin-1 in the PS-vesicles did

not have major effects on the clustering induced by $C_1C_2BMUN$, as shown by the bar diagrams of *Figures 2B–E* and by the corresponding intensity autocorrelation curves (*Figure 2F*). The two types of representations provide different views of the DLS data; below we use one or the other depending on the aspect that we want to emphasize. We note that there is some degree of variability among the particle sizes observed, which makes it difficult to draw firm conclusions from the small differences observed in *Figures 2B–E*. Hence, these data show that PS is the main determinant for the vesicle clustering activity of $C_1C_2BMUN$, although we cannot rule out that synaptobrevin, syntaxin-1, DAG or $PIP_2$ might affect clustering to a small degree. Similarly, $Ca^{2+}$ did not have a major effect on clustering of PS-vesicles by $C_1C_2BMUN$, although it might increase clustering to a small extent (*Figure 2—figure supplement 1*).

These results correlate with the observation that the strong stimulatory activity of $C_1C_2BMUN$ in lipid mixing between V- and T-liposomes does not require $Ca^{2+}$, DAG or $PIP_2$ (*Figures 1B,D*), indicating that this activity arises from its ability to bind simultaneously to two PS-containing membranes in a $Ca^{2+}$-independent manner, thus favoring SNARE complex assembly. This activity might contribute to the role of Munc13s in synaptic vesicle docking and is distinct from the function of Munc13-1 in mediating the transition from the syntaxin-1-Munc18-1 complex to the SNARE complex (*Ma et al., 2011*), but likely potentiates this function in the reconstitutions that include Munc18-1 and NSF-αSNAP by placing the MUN domain near the SNARE-Munc18-1 machinery.

## Simultaneous evaluation of lipid and content mixing

As expected, Syt1 $C_2AB$ could not stimulate lipid mixing between V- and T-liposomes in the presence of NSF-αSNAP because NSF-αSNAP disassemble the syntaxin-1-SNAP-25 t-SNARE complex (*Ma et al., 2013*), but it was surprising that Syt1 $C_2AB$ did not have marked effects on the lipid mixing observed in the presence of Munc13-1 $C_1C_2BMUN$, Munc18-1 and NSF-αSNAP (data not shown; see also below). This observation prompted us to investigate to what extent the lipid mixing observed in these experiments reflects real membrane fusion. For this purpose, we used an assay that simultaneously measures lipid mixing from de-quenching of the fluorescence of Marina Blue-labeled lipids and content mixing from the development of FRET between PhycoE-Biotin trapped in the T-liposomes and Cy5-Streptavidin trapped in the V-liposomes (*Zucchi and Zick, 2011*). Addition of unlabeled streptavidin to the reaction ensures that the observed FRET arises only from content mixing. Using this approach, we again observed that efficient lipid mixing in the presence of NSF-αSNAP required $C_1C_2BMUN$ and Munc18-1, as well as $Ca^{2+}$, and that Syt1 $C_2AB$ did not markedly affect lipid mixing under these conditions (*Figure 3A,C*). However, content mixing in the presence of Munc13-1 $C_1C_2BMUN$, Munc18-1, NSF-αSNAP and $Ca^{2+}$ was inefficient in the absence of Syt1 $C_2AB$ and was strongly enhanced by Syt1 $C_2AB$ (*Figure 3B,D*). The difference between lipid and content mixing in the absence of Syt1 $C_2AB$ emphasizes the fact that lipid mixing may not necessarily reflect true membrane fusion, as described in previous studies [e.g. (*Chan et al., 2009*; *Zick and Wickner, 2014*; *Kyoung et al., 2011*; *Diao et al., 2012*; *Lai et al., 2014*)]. Overall, our results show that Syt1 $C_2AB$ selectively enhances content mixing but not lipid mixing under our conditions. These findings indicate that Syt1 plays a role in membrane fusion, in agreement with results from single vesicle assays using full-length Syt1 (*Kyoung et al., 2011*; *Diao et al., 2012*).

Experiments performed without streptavidin revealed a small amount of leakiness in reactions containing $C_1C_2BMUN$, Munc18-1, NSF-αSNAP and Syt1 $C_2AB$, but the leakiness occurred mostly in the beginning and likely arises because of the presence of a population of small, relatively unstable vesicles (*Figure 3—figure supplement 1*). Note that in these assays much of the Cy5 fluorescence increase caused by FRET from PhycoE (reflecting content mixing) should occur during the first round of fusion and that no further substantial increases are thus expected in subsequent rounds of fusion or upon detergent addition. Correspondingly, the maximum Cy5 fluorescence observed in our most efficient fusion reactions was similar to that observed upon detergent addition (e.g. *Figure 3D*, red curve; see Materials and Methods). In contrast, the lipid mixing signal expressed as percentage of maximum Marina Blue fluorescence is much smaller in the same reactions (e.g. *Figure 3C*, red curve) because fluorescence de-quenching is expected to continue in successive rounds of fusion and to undergo a further, large increase upon detergent addition due to additional probe dilution.

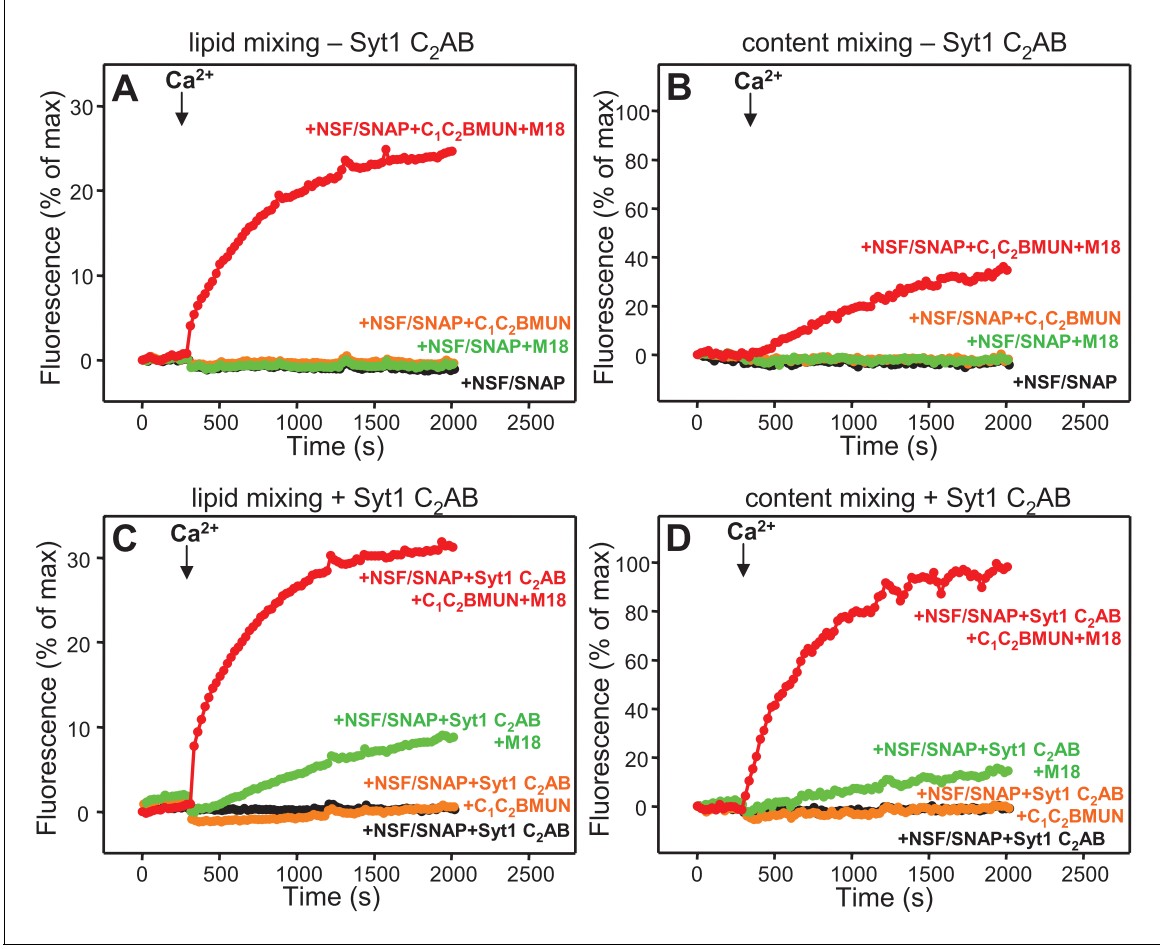

**Figure 3.** Syt1 is required for efficient content mixing but not lipid mixing in reconstitutions including Munc18-1, Munc13-1 $C_1C_2$BMUN and NSF-α SNAP. Lipid mixing (**A,C**) between V- and T-liposomes was measured from the fluorescence de-quenching of Marina Blue-labeled lipids and content mixing (**B,D**) was monitored from the development of FRET between PhycoE-Biotin trapped in the T-liposomes and Cy5-Streptavidin trapped in the V-liposomes. The assays were performed in the presence of different combinations of Munc13-1 $C_1C_2$BMUN, Munc18-1 (M18) and NSF-αSNAP (NSF/SNAP), and in the absence (**A,B**) or presence (**C,D**) of Syt1 $C_2$AB fragment. Experiments were started in the presence of 100 µM EGTA and 5 µM streptavidin, and $Ca^{2+}$ (600 µM) was added after 300 s.

The following figure supplement is available for figure 3:

**Figure supplement 1.** Assessment of leakiness in content mixing assays.

## The Munc13-1 $C_2$C domain strongly stimulates liposome fusion

The $Ca^{2+}$-dependent membrane fusion observed in the presence of $C_1C_2$BMUN, Munc18-1, NSF-α SNAP and Syt1 $C_2$AB is efficient but is much slower than that of neurotransmitter release, suggesting that our reconstitutions lack at least one key factor that contributes to the high speed of release in vivo. We hypothesized that the Munc13-1 $C_2$C domain might be such a factor based on evidence suggesting that this domain plays an important role in release (*Stevens et al., 2005*; *Madison et al., 2005*). To test this hypothesis, we performed fusion assays between V- and T-liposomes, with or without Syt1 $C_2$AB, in the presence of Munc18-1, NSF-αSNAP and fragments of Munc13-1 that contained the MUN domain alone or together with the $C_1C_2$B region, the $C_2$C domain, or both (MUN, $C_1C_2$BMUN, MUN$C_2$C and $C_1C_2$BMUN$C_2$C, respectively). We found that the MUN and MUN$C_2$C fragments did not support membrane fusion but $C_1C_2$BMUN$C_2$C was much more efficient than $C_1C_2$BMUN in facilitating $Ca^{2+}$-dependent fusion; in fact, Syt1 $C_2$AB had no marked effect in the experiments performed with $C_1C_2$BMUN$C_2$C (*Figure 4* and *Figure 4—figure supplement 1*), likely because this fragment is already highly efficient in promoting fusion in the time scale of our

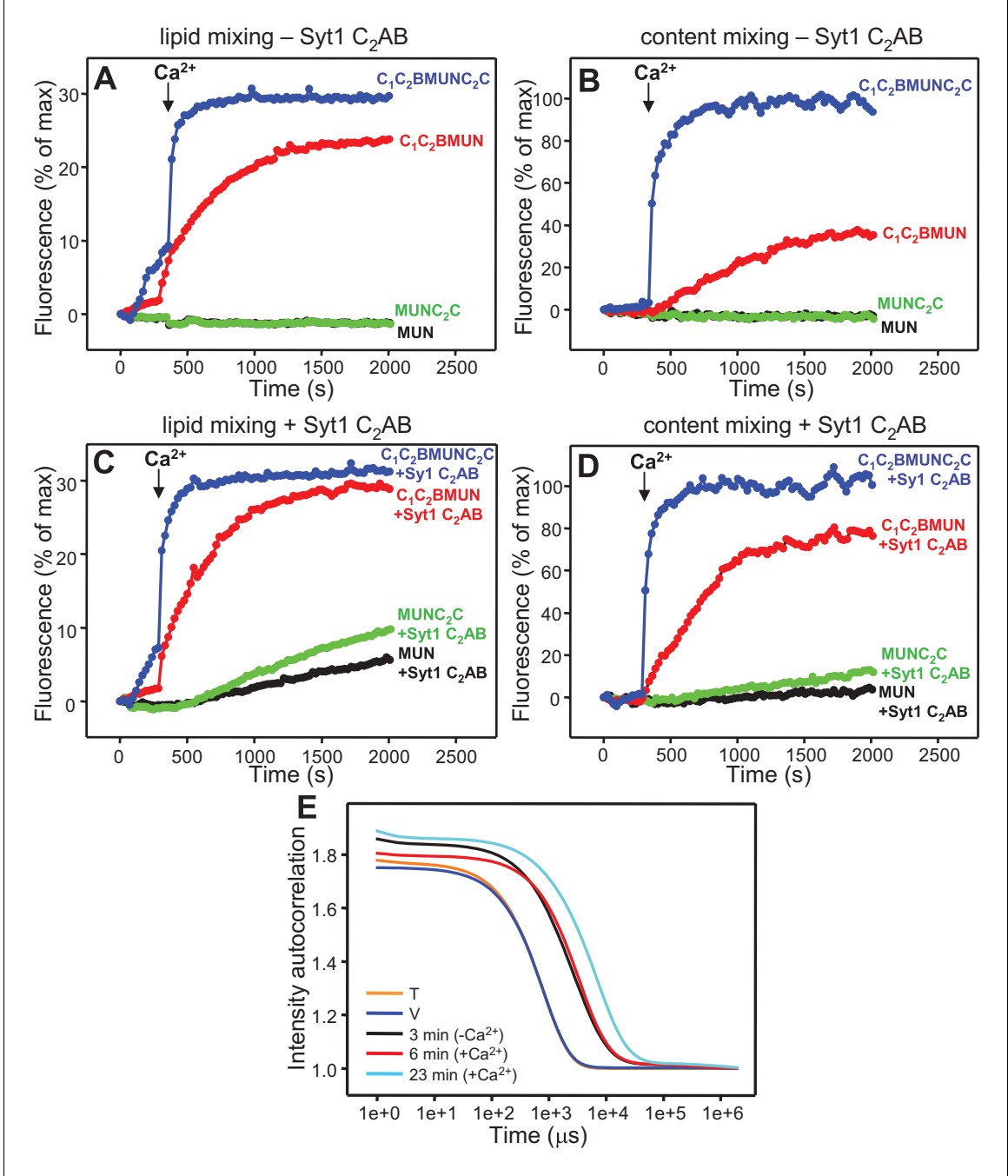

**Figure 4.** The Munc13-1 $C_2C$ domain strongly stimulates membrane fusion. Lipid mixing (**A,C**) between V- and T-liposomes was measured from the fluorescence de-quenching of Marina Blue-labeled lipids and content mixing (**B,D**) was monitored from the development of FRET between PhycoE-Biotin trapped in the T-liposomes and Cy5-Streptavidin trapped in the V-liposomes. The assays were performed in the presence of Munc18-1, NSF-α SNAP and distinct Munc13-1 fragments as indicated, without (**A,B**) or with (**C,D**) Syt1 $C_2AB$ fragment. Experiments were started in the presence of 100 μM EGTA and 5 μM streptavidin, and $Ca^{2+}$ (600 μM) was added after 300 s. (**E**) Intensity autocorrelation curves measured by DLS for isolated V- or T-liposomes, or at different time points as indicated in a fusion reaction performed as in (**A,B**) with $C_1C_2BMUNC_2C$ and 8-fold dilution of all proteins and liposomes. Lipid and content mixing curves for this reaction, as well as particle size distributions corresponding to several of these intensity autocorrelation curves, are shown in *Figure 4—figure supplement 4*.

The following figure supplements are available for figure 4:

**Figure supplement 1.** Quantification of lipid and content mixing experiments of *Figure 4*.

*Figure 4 continued on next page*

*Figure 4 continued*

**Figure supplement 2.** Dependence of lipid and content mixing on DAG and PIP$_2$.

**Figure supplement 3.** Ca$^{2+}$-dependence of membrane fusion.

**Figure supplement 4.** Analysis of particle size during fusion assays between V- and T-liposomes in the presence of Munc18-1, NSF-αSNAP and Munc13-1 C$_1$C$_2$BMUNC$_2$C.

measurements. Note that, in the absence of Ca$^{2+}$, C$_1$C$_2$BMUNC$_2$C was also more active than C$_1$C$_2$BMUN in promoting lipid mixing, but did not stimulate content mixing.

These results show that, indeed, the Munc13-1 C$_2$C domain plays a key role in stimulating liposome fusion, but the lack of activity of the MUNC$_2$C fragment shows that the region spanning the C$_1$ and C$_2$B domains is also important for such stimulation. Since these two domains bind DAG and PIP$_2$, respectively, we tested the effects of removing DAG, PIP$_2$ or both in the T-liposomes in the full reconstitutions including C$_1$C$_2$BMUNC$_2$C and observed considerable impairments of fusion (*Figure 4—figure supplement 2*), similar to those observed in the lipid mixing assays of *Figure 1E*. We also attempted to examine the Ca$^{2+}$-dependence of fusion in these full reconstitutions, which were normally performed with 100 μM EGTA to chelate any residual Ca$^{2+}$ before addition of 600 μM Ca$^{2+}$ (to make the concentration of free Ca$^{2+}$ 500 μM). However, experiments where we added 100 or 120 μM Ca$^{2+}$ at 300 s (*Figure 4—figure supplement 3*) yielded similar fusion efficiency to that observed when we added 600 μM Ca$^{2+}$ (*Figure 4D*). Since the EGTA present should chelate most of the added 100 μM Ca$^{2+}$, these results suggest that a small amount of residual Ca$^{2+}$ (likely in the 1 μM range or below) is sufficient to trigger fusion in these experiments, but further research will be required to assess the Ca$^{2+}$-dependence more accurately. Note that the sensitivity of the reaction to such low Ca$^{2+}$ concentrations, compared to those that activate Syt1 (*Fernandez-Chacon et al., 2001*), can be attributed to the C$_2$B domain present in C$_1$C$_2$BMUNC$_2$C (*Shin et al., 2010*) (see below), and that residual Ca$^{2+}$ might arise from the purified C$_1$C$_2$BMUNC$_2$C fragment, which we did not treat with Ca$^{2+}$ chelators to avoid removal of the Zn$^{2+}$ ions bound to the C$_1$ domain.

We also compared the effects of the Munc13-1 C$_1$C$_2$BMUN and C$_1$C$_2$BMUNC$_2$C fragments on liposome fusion in the absence of NSF-αSNAP. Interestingly, C$_1$C$_2$BMUNC$_2$C alone stimulated fusion between V- and T-liposomes strongly but in a Ca$^{2+}$ independent manner (*Figure 5A,B* and *Figure 5—figure supplement 1A,B*), unlike the reactions that included Munc18-1 and NSF-αSNAP (*Figure 4*). The fusion efficiency caused by C$_1$C$_2$BMUNC$_2$C alone was similar to that induced by Syt1 C$_2$AB alone in the presence of Ca$^{2+}$ and appeared to be somewhat increased by addition of Munc18-1, even though Munc18-1 alone did not stimulate fusion (*Figure 5* and *Figure 5—figure supplement 1*). Syt1 C$_2$AB did not enhance fusion further in the reactions containing Munc18-1 and C$_1$C$_2$BMUNC$_2$C, presumably because fusion is already highly efficient. However, Syt1 C$_2$AB did enhance fusion in the presence of Munc18-1 and C$_1$C$_2$BMUN fragment, which is less efficient than C$_1$C$_2$BMUNC$_2$C (*Figure 5* and *Figure 5—figure supplement 1*).

## Munc13-1 C$_1$C$_2$BMUNC$_2$C favors bridging of V- to T-liposomes

Our results suggest that the Munc13-1 C$_2$C domain contributes strongly to promote membrane fusion but in a Ca$^{2+}$-independent manner, consistent with the fact that it does not contain a full set of the aspartate side chains that commonly form the Ca$^{2+}$-binding sites of C$_2$ domains (*Rizo and Sudhof, 1998*; *Brose et al., 1995*). To investigate the mechanism underlying these findings, we first performed clustering experiments with T- and V-liposomes under the same conditions of *Figures 5A,B* with addition of only C$_1$C$_2$BMUN or C$_1$C$_2$BMUNC$_2$C at different concentrations, and monitored the particle size after 3 min by DLS. Although these measurements reflect not only liposome clustering but also fusion, clustering should dominate the formation of large particles. These data indicated that C$_1$C$_2$BMUNC$_2$C is somewhat more efficient in liposome clustering than C$_1$C$_2$BMUN, exhibiting substantial clustering activity at 50–100 nM concentration (*Figure 6—figure supplement 1*). Liposome co-floatation assays also suggested that the presence of the C$_2$C domain might increase the affinity of C$_1$C$_2$BMUNC$_2$C for liposomes, but it is not sufficient for stable

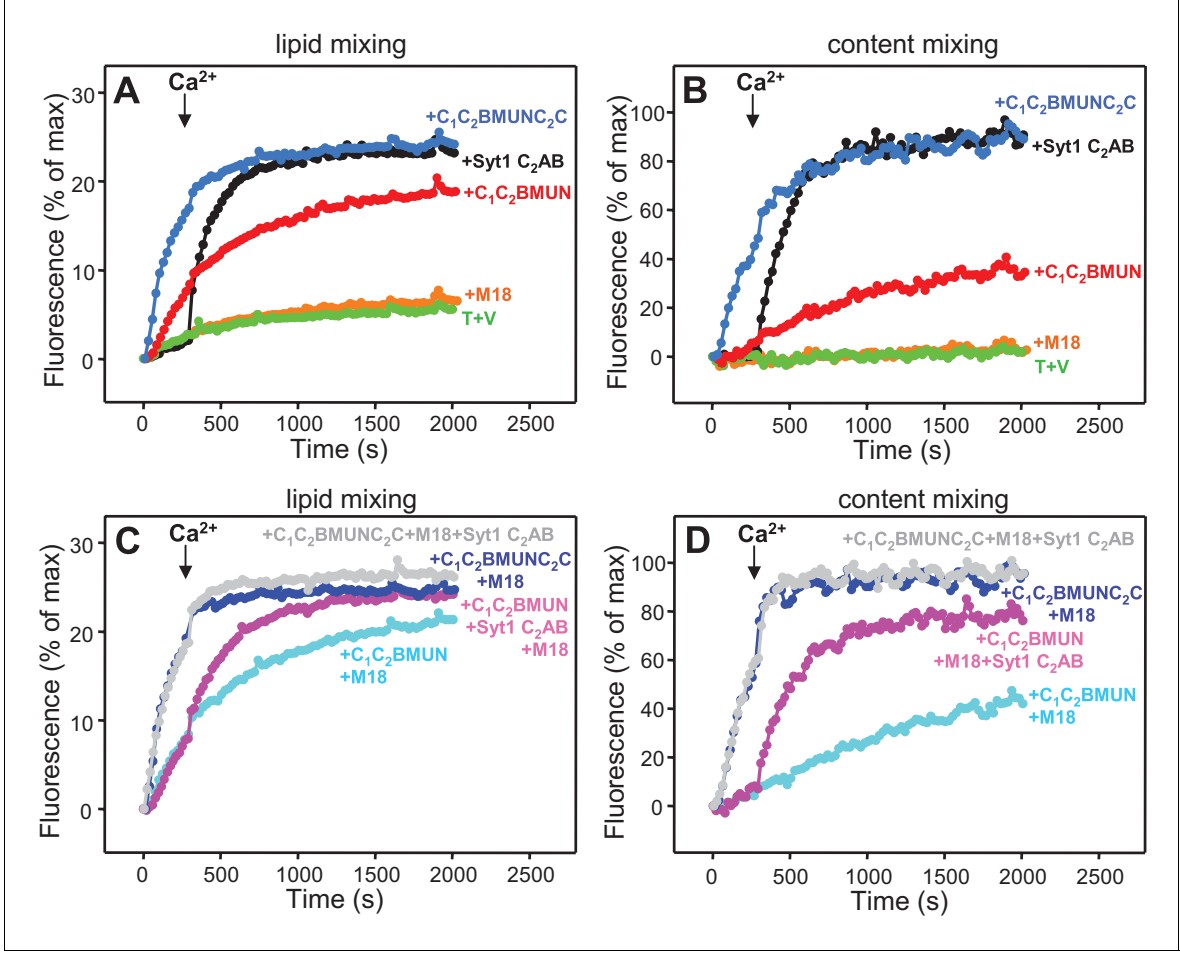

**Figure 5.** Munc13-1 $C_1C_2BMUNC_2C$ can induce $Ca^{2+}$-independent fusion of V- and T-liposomes in the absence of NSF-αSNAP. Lipid mixing (**A,C**) between V- and T-liposomes was measured from the fluorescence de-quenching of Marina Blue-labeled lipids and content mixing (**B,D**) was monitored from the development of FRET between PhycoE-Biotin trapped in the T-liposomes and Cy5-Streptavidin trapped in the V-liposomes. The assays were performed in the presence of different combinations of Munc18-1 (M18), Syt1 $C_2AB$ fragment and Munc13-1 $C_1C_2BMUN$ or $C_1C_2BMUNC_2C$ as indicated. Experiments were started in the presence of 100 µM EGTA and 5 µM streptavidin, and $Ca^{2+}$ (600 µM) was added after 300 s.

The following figure supplement is available for figure 5:

**Figure supplement 1.** Quantification of lipid and content mixing experiments of **Figure 5**.

liposome binding in the context of $MUNC_2C$ (**Figure 6—figure supplement 2**). However, it was unclear whether these effects of the $C_2C$ domain on clustering and liposome affinity are sufficient to explain those observed on membrane fusion.

Interestingly, when we analyzed clustering of V- and T-liposomes separately, we found that $C_1C_2BMUNC_2C$ clustered V-liposomes only in the presence of $Ca^{2+}$ whereas it clustered T-liposomes in the absence and presence of $Ca^{2+}$ (**Figure 6—figure supplement 3**). These results contrast with those obtained with $C_1C_2BMUN$, which clusters V-liposomes even in the absence of $Ca^{2+}$ (**Figure 2**), and suggest a delicate interplay between multiple lipid binding sites within these large protein fragments (see discussion). Since $C_1C_2BMUNC_2C$ clustered mixtures of T- and V-liposomes more efficiently than $C_1C_2BMUN$ (**Figure 6—figure supplement 1**), these data suggested that $C_1C_2BMUNC_2C$ may preferentially bridge V-liposomes to T-liposomes. To test this notion more rigorously, we analyzed liposome clustering after 3 min at 20°C to minimize contributions of liposome fusion to the DLS data. Lipid mixing assays confirmed that very little fusion occurs under these conditions (**Figure 6—figure supplement 4**). At 20°C, $C_1C_2BMUNC_2C$ again was able to cluster T-liposomes but not V-liposomes in the absence of $Ca^{2+}$, which required $Ca^{2+}$ for

clustering (*Figures 6A–D*). To test whether populations of clustered and non-clustered liposomes can be distinguished by DLS, we used plain vesicles that did not contain PS or proteins and that, correspondingly, were not clustered by $C_1C_2BMUNC_2C$ (*Figure 6E*). Indeed, a clearly bimodal distribution of clustered and non-clustered liposomes was observed by DLS when we added $C_1C_2BMUNC_2C$ to a 1:1 mixture of plain liposomes and T-liposomes (*Figure 6F*). Importantly, only clustered vesicles were detectable by DLS analysis of a 1:1 mixture of V- and T-liposomes in the presence of $C_1C_2BMUNC_2C$ and the absence of $Ca^{2+}$ (*Figure 6G,H*). These data show that, whereas $Ca^{2+}$-free $C_1C_2BMUNC_2C$ does not cluster V-liposomes (*Figure 6A,B*), it can bridge V-liposomes to T-liposomes.

Overall, this analysis suggests that the dramatic stimulation of membrane fusion caused by $C_1C_2BMUNC_2C$ compared to $C_1C_2BMUN$ (*Figures 4,5*) arises at least in part because of this preferential bridging of V- to T-liposomes by $C_1C_2BMUNC_2C$, and perhaps because such bridging is more efficient (*Figure 6—figure supplement 1*) and/or longer lasting. Hence, these results further suggest that Munc13s play a role in bridging synaptic vesicles to the plasma membrane, which may contribute to its function in docking and facilitate the activity of the MUN in opening syntaxin-1 (*Figure 7*), and indicate that this bridging activity involves a synergy between the $C_1$, $C_2B$ and $C_2C$ domains. However, while the ability of $Ca^{2+}$-free $C_1C_2BMUNC_2C$ to bridge V-liposomes to T-liposomes explains its stimulation of fusion between these liposomes in the absence of $Ca^{2+}$ (*Figure 5*), it is unclear why the dramatic stimulation of fusion caused by $C_1C_2BMUNC_2C$ in the presence of Munc18-1 and NSF-αSNAP requires $Ca^{2+}$ (*Figure 4B*).

Attempts to test whether $Ca^{2+}$ causes substantial additional clustering under these conditions were hindered by saturation of the DLS detector as the reaction progressed. We thus performed additional lipid and content mixing assays with all protein and liposome concentrations diluted 2-fold, 4-fold and 8-fold, and found that $Ca^{2+}$ still stimulated fusion strongly even in the most diluted conditions, albeit at a somewhat slower rate (*Figure 4—figure supplement 4A,B*). Analysis of the fusion reaction with the 8-fold dilution by DLS revealed that efficient clustering already occurred after 3 min in the absence of $Ca^{2+}$, and the particle size increased to a very little extent one minute after $Ca^{2+}$ addition (*Figure 4E*; *Figure 4—figure supplement 4C–F*), although the size did increase considerably as the reaction progressed to completion. These results suggest that the action of $C_1C_2BMUNC_2C$ in these experiments is not limited to bridging V- to T-liposomes but also involves activities downstream of such bridging.

## Incorporation of membrane-anchored Syt1 in the reconstitution assays

The use of the soluble Syt1 $C_2AB$ fragment in our assays allows analysis of the effects of including or omitting the Syt1 $C_2$ domains on fusion, but in vivo Syt1 is anchored on synaptic vesicles. To study how membrane-anchoring of Syt1 affects fusion in our assays, we used liposomes containing synaptobrevin and a Syt1 fragment spanning its transmembrane (TM) and cytoplasmic regions (VSyt1-liposomes). These liposomes contained a smaller percentage of PS (6.8%) that resembles that of synaptic vesicles and prevents inhibition of fusion due to binding of Syt1 to the membrane where it is anchored (*Stein et al., 2007*). The VSyt1-liposomes fused efficiently with T-liposomes in a $Ca^{2+}$-independent manner, as described previously (*Stein et al., 2007*), and the Munc13-1 $C_1C_2BMUN$ fragment slightly increased the fusion efficiency, but Munc18-1 had no marked effect (*Figure 8A,B*). However, as expected, fusion was abolished by NSF-αSNAP and, in their presence, fusion required Munc18-1, $C_1C_2BMUN$ and $Ca^{2+}$ (*Figure 8C,D*). These latter results are similar to those obtained in the experiments where the soluble Syt1 $C_2AB$ was added instead of incorporating Syt1 into the V-liposomes (*Figure 3C,D*).

Fusion assays between VSyt1- and T-liposomes in the presence of NSF-αSNAP, Munc18-1 and different Munc13-1 fragments (*Figure 9A,B* and *Figure 9—figure supplement 1*) also yielded similar results to those obtained with soluble Syt1 $C_2AB$ (*Figure 4C,D*), as $C_1C_2BMUNC_2C$ was much more active than $C_1C_2BMUN$ while MUN and $MUNC_2C$ remained inactive. Content mixing between VSyt1- and T-liposomes in the presence of NSF-αSNAP, Munc18-1 and $C_1C_2BMUNC_2C$ again required $Ca^{2+}$, even though there was lipid mixing before adding $Ca^{2+}$, and was very fast upon $Ca^{2+}$ addition. Moreover, absence of either Munc18-1 or Munc13-1 $C_1C_2BMUNC_2C$ completely abolished fusion (*Figure 9C,D*), in correlation with the absolute requirement of both proteins for neurotransmitter release in vivo.

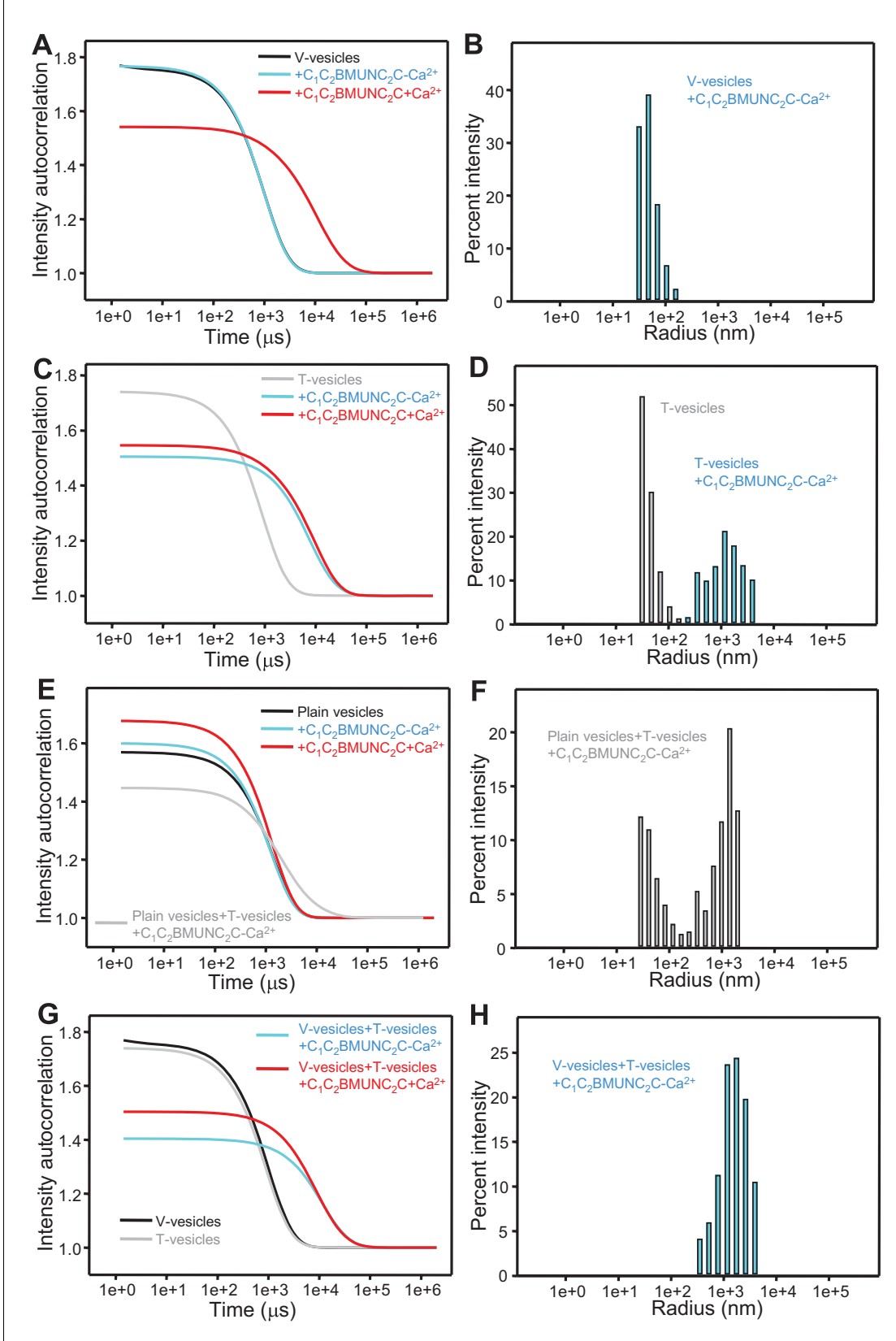

**Figure 6.** The Munc13-1 $C_1C_2BMUNC_2C$ fragment bridges V-liposomes to T-liposomes. (**A,C,E,G**) Intensity autocorrelation curves measured by DLS after 3 min incubations at 20°C on samples containing: (**A**) V-vesicles alone or in the presence of $C_1C_2BMUNC_2C$ and 100 µM EGTA or 500 µM $Ca^{2+}$; (**C**)

*Figure 6 continued on next page*

*Figure 6 continued*

T-vesicles alone or in the presence of $C_1C_2BMUNC_2C$ and 100 µM EGTA or 500 µM $Ca^{2+}$; (**E**) plain vesicles containing no PS alone or in the presence of $C_1C_2BMUNC_2C$ and 100 µM EGTA or 500 µM $Ca^{2+}$, or a 1:1 mixture of plain vesicles and T-vesicles in the presence of $C_1C_2BMUNC_2C$ and 100 µM EGTA; (**G**) V-vesicles alone, T-vesicles alone, or 1:1 mixtures of V- and T-vesicles in the presence of $C_1C_2BMUNC_2C$ and 100 µM EGTA or 500 µM $Ca^{2+}$. (**B,D,F,H**) Bar diagrams showing the particle size distribution in samples containing: (**B**) V-vesicles in the presence of $C_1C_2BMUNC_2C$ and 100 µM EGTA; (**D**) T-vesicles alone or in the presence of $C_1C_2BMUNC_2C$ and 100 µM EGTA; (**F**) a 1:1 mixture of plain vesicles and T-vesicles in the presence of $C_1C_2BMUNC_2C$ and 100 µM EGTA; (**H**) a 1:1 mixture of V- and T-vesicles in the presence of $C_1C_2BMUNC_2C$ and 100 µM EGTA. These bar diagrams correspond to the autocorrelation curves of selected samples among those shown in (**A,C,E,G**) and are intended to illustrate that mixtures of clustered and non-clustered vesicles can be readily distinguished (**F**), and that $Ca^{2+}$-free $C_1C_2BMUNC_2C$ does not cluster isolated V-vesicles (**B**) but bridges V- to T-vesicles (**H**).

The following figure supplements are available for figure 6:

**Figure supplement 1.** Concentration dependence of the liposome clustering activity of Munc13-1 $C_1C_2BMUN$ and $C_1C_2BMUNC_2C$.

**Figure supplement 2.** Lipid binding to distinct Munc13-1 fragments monitored by liposome co-floatation assays.

**Figure supplement 3.** $Ca^{2+}$-free $C_1C_2BMUNC_2C$ does not cluster V-liposomes.

**Figure supplement 4.** Minimal stimulation of lipid mixing between V- and T-liposomes in the presence of $C_1C_2BMUNC_2C$ at 20°C.

## Functional importance of $Ca^{2+}$ binding to the Munc13-1 $C_2B$ domain

The finding that the data obtained with NSF-αSNAP, Munc18-1 and Munc13-1 $C_1C_2BMUNC_2C$ do not depend strongly on Syt1 but exhibit a drastic $Ca^{2+}$ dependence indicates that such $Ca^{2+}$ dependence arises from the Munc13-1 $C_2B$ domain, which contains the only known $Ca^{2+}$-binding sites in these proteins (*Shin et al., 2010*). To test this idea, we analyzed fusion between VSyt1- and T-liposomes in the presence of NSF-αSNAP, Munc18-1 and a mutant Munc13-1 $C_1C_2BMUNC_2C$ fragment where two of the aspartate $Ca^{2+}$ ligands were mutated to asparagine (D705N,D711N) to disrupt $Ca^{2+}$ binding. We found that content mixing was strongly impaired by the D706N,D711N mutation while $Ca^{2+}$-independent lipid mixing was not affected (*Figure 9E,F* and *Figure 9—figure supplement 2*), indicating that $Ca^{2+}$ binding to the Munc13-1 $C_2B$ domain is critical for fusion under these conditions. To examine how these results are related to the membrane bridging activity of $C_1C_2BMUNC_2C$, we analyzed the particle size in fusion reactions where all protein and liposome concentrations were diluted eight-fold, which still allows a strong $Ca^{2+}$-induced stimulation of content mixing for WT $C_1C_2BMUNC_2C$ (as observed in the experiments performed with the soluble Syt1 $C_2AB$ fragment; *Figure 4—figure supplement 4A,B*) but not for the D706N,D711N mutant (*Figure 9—figure supplement 3A–D*). DLS analysis of the 8-fold diluted fusion reactions including WT $C_1C_2BMUNC_2C$ showed that much of the liposome clustering had already occurred after 3 min in the absence of $Ca^{2+}$, while little changes were observed 1 min after adding $Ca^{2+}$ and a moderate increase in particle size occurred as the reaction progressed to completion (*Figure 9—figure supplement 3E*). In analogous reactions with the $C_1C_2BMUNC_2C$ D706N,D711N mutant, efficient clustering occurred after 3 min and did not increase further afterwards (*Figure 9—figure supplement 3F*).

These results suggest that, while $Ca^{2+}$ binding to the $C_2B$ domain of WT $C_1C_2BMUNC_2C$ might contribute to more efficient bridging of V- to T-liposomes, it is unlikely that an effect on such bridging alone can explain the dramatic enhancement of content mixing induced by $Ca^{2+}$. Note also that the finding that the D706N,D711N mutation disrupts content mixing but not $Ca^{2+}$-independent lipid mixing (*Figure 9E*) correlates with the observation that efficient content mixing in the presence of WT $C_1C_2BMUNC_2C$ requires $Ca^{2+}$, while there is substantial lipid mixing without $Ca^{2+}$ (*Figures 9A, C,E*). Indeed, quantification at 300 s, before $Ca^{2+}$ addition, showed that the fluorescence increase reflecting lipid mixing was 28.9% of that observed 200 s after $Ca^{2+}$ addition while content mixing was minimal at 300 s (fluorescence 3.3% of that observed 200 s after adding $Ca^{2+}$) (*Figure 9—figure supplement 2*). This difference is exacerbated by the fact that the maximal fluorescence associated with content mixing is expected to correspond to only one round of fusion, whereas additional rounds of fusion can contribute to the maximal fluorescence in the lipid mixing signal. Hence, these results suggest that $C_1C_2BMUNC_2C$, NSF-αSNAP and Munc18-1 enable formation of a 'primed

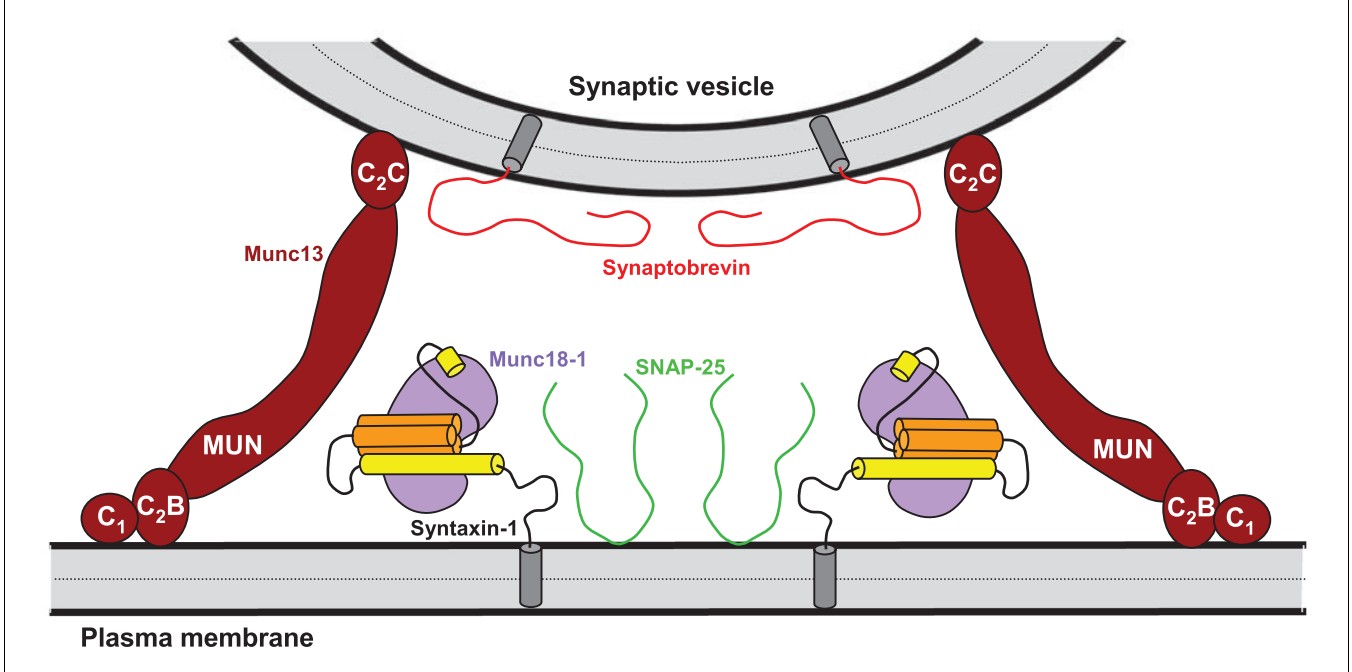

**Figure 7.** Model of how bridging of synaptic vesicles to the plasma membrane by the highly conserved C-terminal region of Munc13s can create a cage-like environment and facilitate the activity of the MUN domain in promoting the transition from the syntaxin-1-Munc18-1 complex to the SNARE complex, thus favoring SNARE complex assembly. Syntaxin-1 ($H_{abc}$ domain, orange; SNARE motif and N-terminus, yellow) is shown in a closed conformation bound to Munc18-1 (purple). Synaptobrevin is shown in red, SNAP-25 in green and the C-terminal region of Munc13-1 in brown. The model is inspired by the ability of $C_1C_2BMUNC_2C$ to bridge V- to T-liposomes (**Figure 6**) and assumes that the $C_1$-$C_2B$ region binds to the plasma membrane while the $C_2C$ domain binds to the vesicle membrane. See text and the legend of **Figure 7—figure supplement 1** for additional details.

The following figure supplement is available for figure 7:

**Figure supplement 1.** Speculative models of membrane bridging by $C_1C_2BMUN$ and $C_1C_2BMUNC_2C$.

state' that is ready for membrane fusion and includes assembled trans-SNARE complexes, as lipid mixing can occur, but requires $Ca^{2+}$ binding to the Munc13-1 $C_2B$ domain for fast full fusion that might be further accelerated by $Ca^{2+}$ binding to Syt1.

## The Munc13-1 $C_2B$ and $C_2C$ domains are critical for neurotransmitter release

Rescue experiments in which Munc13-1 fragments were overexpressed using the Semliki Forest virus in neurons from Munc13-1/2 double KO mice indicated that the MUN domain is sufficient to rescue neurotransmitter release (**Basu et al., 2005**), but other functional studies in chromaffin cells and C. elegans indicated that the $C_2C$ domain is also critical to rescue Munc13 function (**Stevens et al., 2005**; **Madison et al., 2005**). To clarify the functional importance of different domains for Munc13-1, we performed additional rescue experiments in autaptic neuronal cultures from Munc13-1/2 double KO mice (**Varoqueaux et al., 2002**) but expressing Munc13-1 fragments with a lentiviral expression vector, which does not overexpress proteins at such high levels as the Semliki Forest virus. Analysis of excitatory postsynaptic currents (EPSCs) revealed that a Munc13-1 fragment encompassing the $C_1C_2BMUNC_2C$ region rescued close to 50% of the EPSC amplitude compared to rescue with WT Munc13-1, whereas Munc13-1 fragments spanning the MUN, $MUNC_2C$ or $C_1C_2BMUN$ sequences led to only very small levels of rescue (**Figure 10A,B**). Similar results were obtained when we analyzed the readily-releasable pool using hypertonic sucrose (**Figure 10C,D**). These results need to be examined with caution because distinct expression levels were observed for the different fragments (**Figure 10—figure supplement 1**). However, all Munc13-1 fragments were expressed at higher levels than WT, and hence their levels should be sufficient to rescue release if the fragments are functional. Indeed,

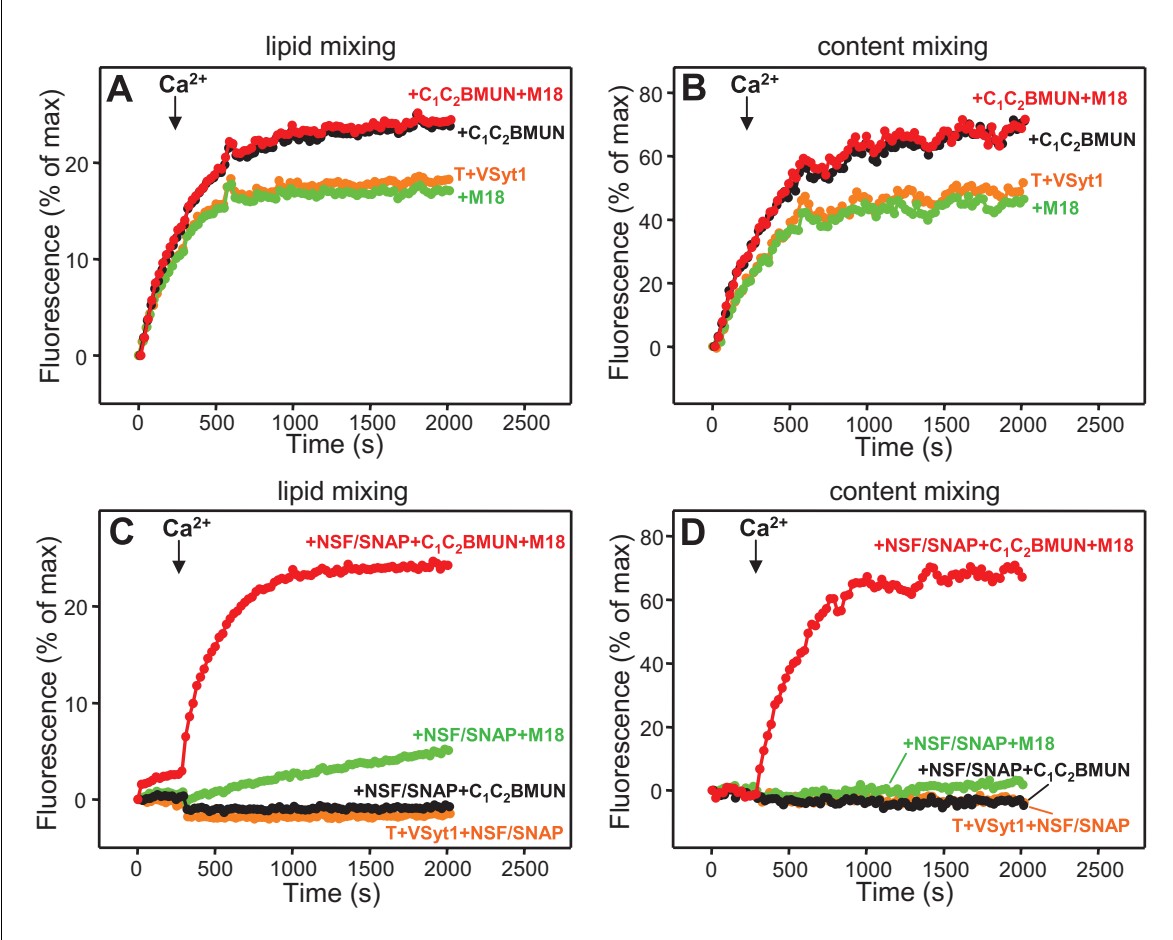

**Figure 8.** Ca²⁺-independent membrane fusion between Syt1-containing V-liposomes and T-liposomes becomes Ca²⁺-dependent in the presence of Munc18-1, Munc13-1 $C_1C_2$BMUN and NSF-αSNAP. Lipid mixing (**A,C**) between V-liposomes containing Syt1 (VSyt1) and T-liposomes was measured from the fluorescence de-quenching of Marina Blue-labeled lipids and content mixing (**B,D**) was monitored from the development of FRET between PhycoE-Biotin trapped in the T-liposomes and Cy5-Streptavidin trapped in the V-liposomes. The assays were performed in the presence of different combinations of Munc18-1 (M18), Munc13-1 $C_1C_2$BMUN and NSF-αSNAP as indicated. Experiments were started in the presence of 100 μM EGTA and 5 μM streptavidin, and Ca²⁺ (600 μM) was added after 300 s.

since there is no release in Munc13-1/2 double KO neurons (*Varoqueaux et al., 2002*), the very small amounts of release observed for the rescues with MUN, MUNC₂C and $C_1C_2$BMUN fragments imply that these fragments can perform Munc13-1 function to some degree, albeit with very low efficiency, and vast protein production may have compensated for functional deficiency in the previous rescues with Semliki Forest virus overexpression of the MUN domain (*Basu et al., 2005*). Importantly, the robust rescue observed with the $C_1C_2$BMUNC₂C fragment compared to $C_1C_2$BMUN (*Figure 10*) shows that the Munc13-1 C₂C domain indeed plays an important role in neurotransmitter release, in clear correlation with our reconstitution data. Moreover, the $C_1C_2$B region is also critical for Munc13-1 function, as the MUNC₂C fragment is much less active than $C_1C_2$BMUNC₂C in both the rescue experiments (*Figure 10*) and in our fusion assays (*Figures 4* and *9*).

## Discussion

Great advances have been made to characterize the central components of the neurotransmitter release machinery, but fundamental questions remain about how these components work together to trigger Ca²⁺-dependent membrane fusion. Strong evidence indicates that Munc18-1 and Munc13s orchestrate SNARE complex formation in an NSF-SNAP-resistant manner (*Ma et al., 2013*), and that the participation of Munc13s in SNARE complex formation underlies their key

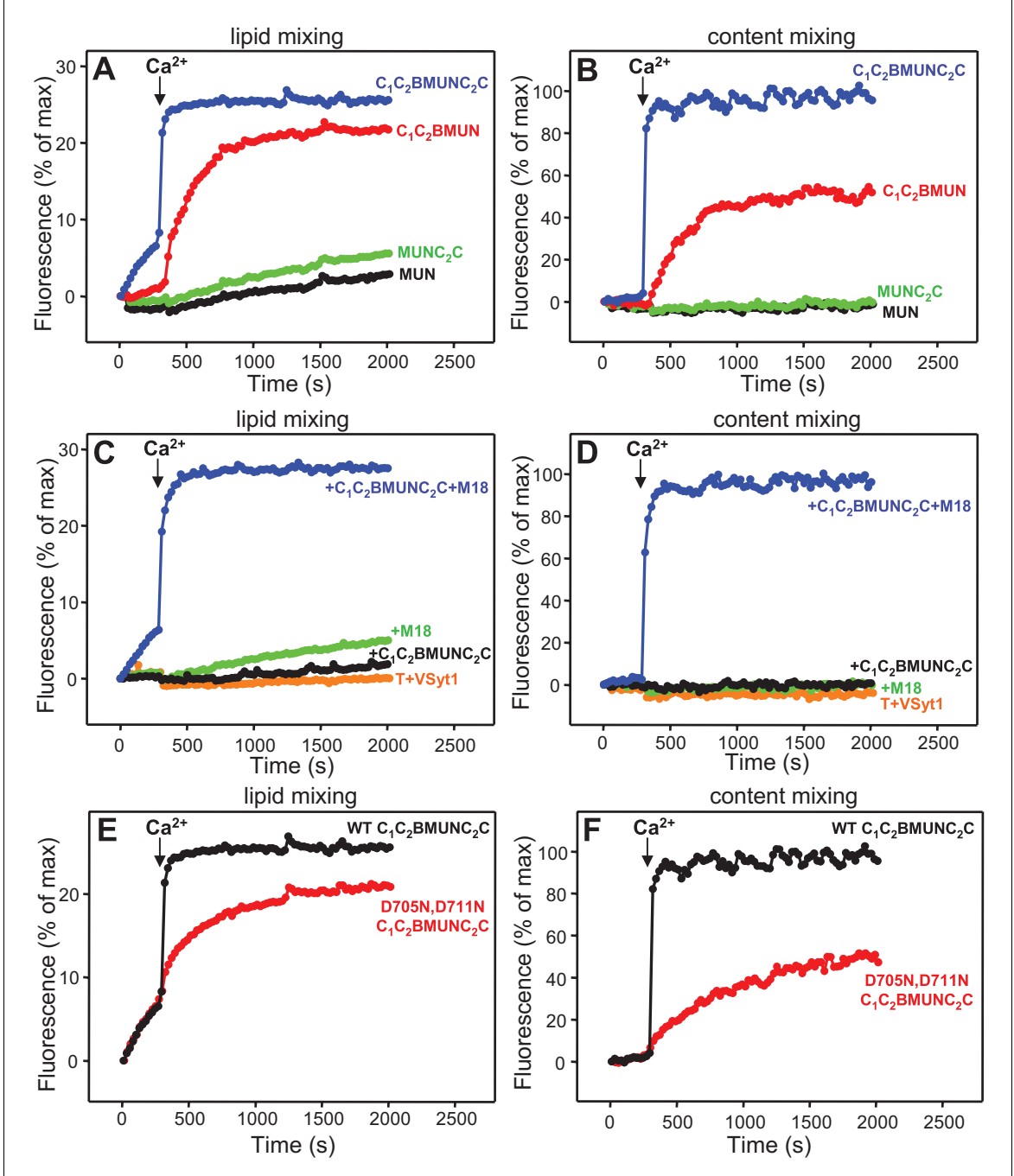

**Figure 9.** Fast, $Ca^{2+}$-dependent membrane fusion between VSyt1- and T-liposomes in the presence of Munc18-1, Munc13-1 $C_1C_2BMUNC_2C$ and NSF-α SNAP, which depends on $Ca^{2+}$ binding to the Munc13-1 $C_2B$ domain. Lipid mixing (**A,C,E**) between V-liposomes containing Syt1 (VSyt1) and T-liposomes was measured from the fluorescence de-quenching of Marina Blue-labeled lipids and content mixing (**B,D,F**) was monitored from the development of FRET between PhycoE-Biotin trapped in the T-liposomes and Cy5-Streptavidin trapped in the V-liposomes. In (**A,B**), the assays were performed in the presence of Munc18-1, NSF-αSNAP and distinct Munc13-1 fragments as indicated. In (**C,D**), experiments were performed in the presence of NSF-αSNAP with or without addition of Munc18-1 and/or Munc13-1 $C_1C_2BMUNC_2C$. In (**E,F**), assays were performed in the presence of Munc18-1, NSF-αSNAP and WT or D705N,D711N mutant Munc13-1 $C_1C_2BMUNC_2C$. All experiments were started in the presence of 100 μM EGTA and 5 μM streptavidin, and $Ca^{2+}$ (600 μM) was added after 300 s.

The following figure supplements are available for figure 9:

**Figure supplement 1.** Quantification of lipid and content mixing experiments of *Figure 9A,B*.

*Figure 9 continued on next page*

*Figure 9 continued*

**Figure supplement 2.** Quantification of lipid and content mixing experiments of *Figure 9E,F*.

**Figure supplement 3.** Analysis of particle size during fusion assays between VSyt1- and T-liposomes in the presence of Munc18-1, NSF-αSNAP and Munc13-1 $C_1C_2BMUNC_2C$.

functions in vesicle docking and priming (*Weimer et al., 2006*; *Hammarlund et al., 2007*; *Imig et al., 2014*). However, it was unclear whether Munc13s have additional roles upstream and/or downstream of SNARE complex assembly. Moreover, it was unknown how the functions of the different domains that form the highly conserved C-terminal region of Munc13s are integrated. Our results now show that both the $C_1C_2B$-region preceding the MUN domain and the $C_2C$ domain at the C-terminus are critical for neurotransmitter release, and suggest a strong functional synergy between these domains that arises because they help bridging the synaptic vesicle and plasma membranes, facilitating the activity of the MUN domain in mediating opening of syntaxin-1 (*Figure 7*). Our results also indicate that the neuronal SNAREs, Munc18-1, Munc13, NSF and α-SNAP are crucial to generate a 'primed state' that includes Munc13 as an integral component and is ready to release but needs $Ca^{2+}$ to trigger fast membrane fusion.

Our reconstitutions recapitulate many key features of synaptic vesicle fusion, providing an ideal system to investigate the mechanism of release. The total abrogation of neurotransmitter release observed in the absence of Munc18-1 and Munc13s (*Verhage et al., 2000*; *Varoqueaux et al., 2002*) established that these two proteins are the most central factors of the membrane fusion apparatus together with the SNAREs; accordingly, membrane fusion depends strictly on Munc18-1 and Munc13-1 $C_1C_2BMUNC_2C$ in our reconstitutions, as shown in a compelling fashion by *Figure 9D*. Moreover, fusion exhibits a tight dependence on $Ca^{2+}$ (*Figure 9D*) and removal of the $C_1$-$C_2B$ region or the $C_2C$ domain of Munc13-1 markedly impairs fusion in our reconstitutions (*Figures 4B,D* and *9B*) as well as neurotransmitter release in neurons (*Figure 10*). The dependence of fusion on DAG and $PIP_2$ (*Figure 4—figure supplement 2*) correlates with the notion that these factors enhance neurotransmitter release in part via their respective interactions with the Munc13 $C_1$ and $C_2B$ domains (*Rhee et al., 2002*; *Shin et al., 2010*). NSF and αSNAP are key for the strict requirement of Munc18-1, Munc13-1 and $Ca^{2+}$ for fusion (*Figures 4,5*), and for the dependence of lipid mixing on DAG and $PIP_2$ (*Figure 1D,E*), because they disassemble syntaxin-1-SNAP-25 heterodimers (*Weber et al., 2000*), ensuring that vesicle docking, priming and fusion proceed through the Munc18-1-Munc13-dependening pathway (*Ma et al., 2013*).

Although our reconstitutions are incomplete (see below), these multiple correlations with physiological data imply that the mechanisms of action of the proteins included are likely to be related at least in part to those operating in vivo. The finding that the Mun13-1 $C_1C_2BMUNC_2C$ fragment can bridge V-liposomes to T-liposomes (*Figure 6*) is particularly revealing because it suggests a natural model for how the different domains that form the conserved C-terminal region of Munc13s cooperate to mediate synaptic vesicle docking and priming (*Figure 7*), providing an explanation for why this fragment rescues neurotransmitter release and stimulates liposome fusion much more efficiently than shorter fragments (*Figures 4*, *9* and *10*). In this model, we assume that NSF-αSNAP disassemble the syntaxin-1-SNAP-25 complex in the T-liposomes and Munc18-1 binds to the released syntaxin-1 folded into a closed conformation. Bridging of the two membranes through respective interactions with the $C_1$-$C_2B$ region and the $C_2C$ domain at opposite ends of the highly elongated MUN domain creates a 'cage-like' environment to facilitate SNARE complex assembly, placing the MUN domain in an ideal position to exert its activity in accelerating the transition from the syntaxin-1-Munc18-1 complex to the SNARE complex (*Figure 7*). This model is consistent with studies that revealed an important function for Munc13s in docking using stringent vesicle-plasma membrane distance criteria [direct contact or <5 nm; (*Weimer et al., 2006*; *Hammarlund et al., 2007*; *Imig et al., 2014*)], and that suggested that docking is equivalent to priming, reflecting partial SNARE complex formation (note that this notion may not be valid for more relaxed definitions of docking). Our model postulates that Munc13s function in docking and priming not only because

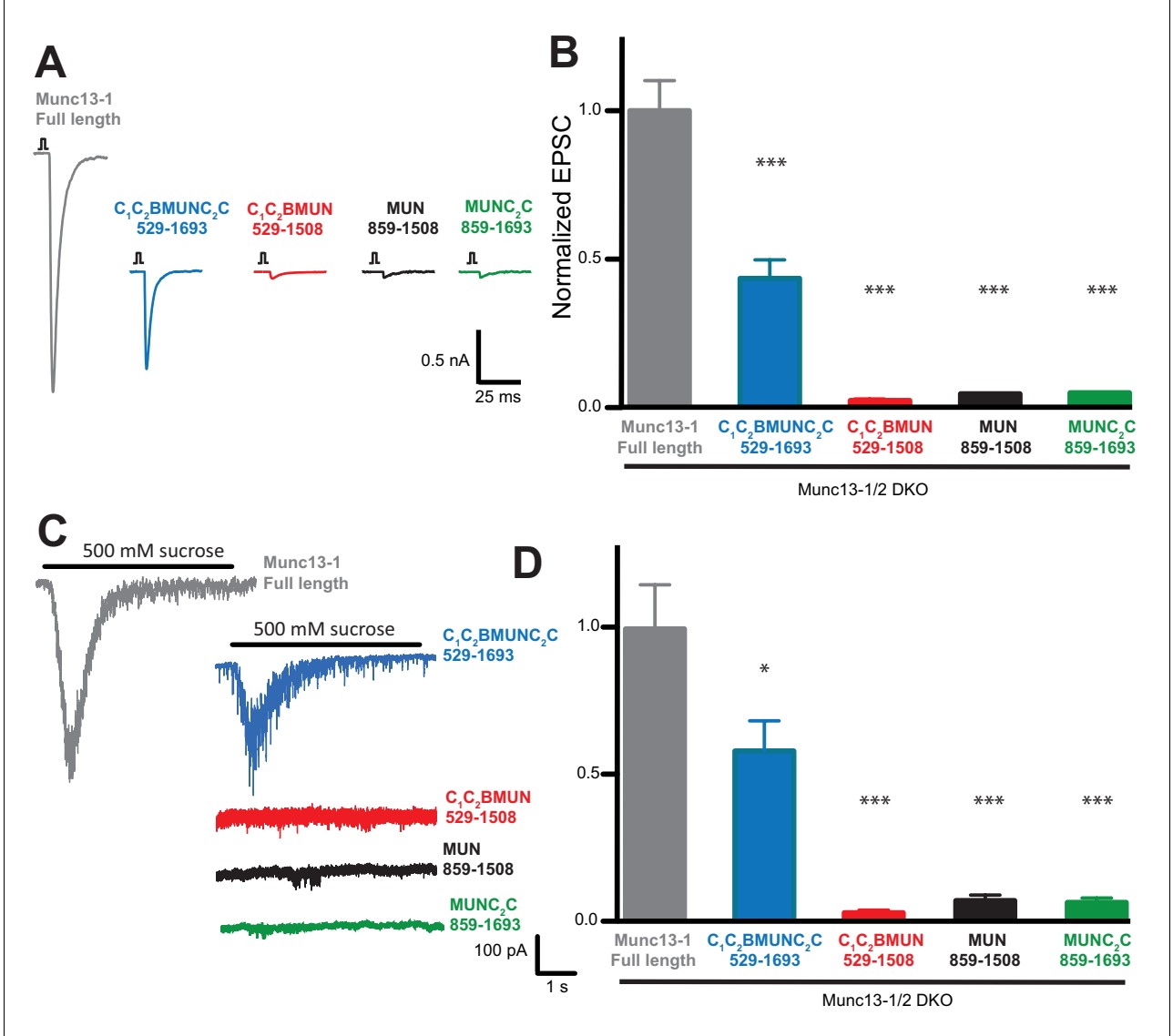

**Figure 10.** The Munc13-1 $C_1$, $C_2B$ and $C_2C$ domains are critical for neurotransmitter release. (**A**) Representatives traces of single AP-evoked EPSCs from Munc13-1/2 DKO hippocampal neurons rescued with Munc13-1 full length, or C-terminal Munc13-1 fragments, in response to 2 ms somatic depolarization. Depolarization artifacts and action potentials were blanked. (**B**) Normalized summary plot of EPSC peak amplitudes from Munc13-1/2 DKO hippocampal neurons rescued with Munc13-1 full length or C-terminal fragments. Data were collected during 4 consecutive days of recording. Data were normalized to the mean value of the control group (Munc13-1 full length). Error bars represent SEM. Normalized data were pooled from two independent cultures. Values that differ significantly from controls are indicated (*p<0.05; ***p<0.001) by Non parametric Kruskal-Wallis test with a *post hoc* Dunn's Multiple comparison test. (**C**) Representative traces of RRP sizes induced by 5 s hypertonic sucrose solution application, from Munc13-1/2 DKO hippocampal neurons rescued with Munc13-1 full length and C-terminal fragments. (**D**) Normalized summary plot of RRP charge. For the rescue experiments, approximately equal *numbers* of green positive Munc13-1/2 DKO neuron rescues with Munc13-1 full length or C-terminal fragments were collected the same day. But due to the fact that neurons that lacked both Munc13-1 and Munc13-2 proteins show no evoked excitatory postsynaptic currents (EPSCs), and no response with sucrose stimulation, EPSC and RRP that show no responses were not quantified in the plots. The following numbers of EPSC or RRP responses were observed out of the total green positive neurons for each condition: FL, 53/55; $C_1C_2BMUNC_2C$, 45/54; $C_1C_2BMUN$, 6/50; MUN, 6/50; $MUNC_2C$, 4/51. EPSC means ± SEM (nA) excluding 0: FL, 1.963 ± 0.2511; $C_1C_2BMUNC_2C$, 0.83120 ± 0.1427; $C_1C_2BMUN$, 0.03298 ± 0.008110; MUN, 0.05984 ± 0.0009355; $MUNC_2C$, 0.05892 ± 0.0009125. EPSC charge means ± SEM (pC) excluding 0: FL, 226.3 ± 42.04; $C_1C_2BMUNC_2C$, 139.5 ± 23.90; $C_1C_2BMUN$, 5.090 ± 1.190; MUN, 1.847 ± 1.092; $MUNC_2C$ 8.259 ± 1.092.

The following figure supplement is available for figure 10:

**Figure supplement 1.** Protein expression from Munc13-1/2 DKO hippocampal mass cultures infected with Munc13-1-Flag and C-terminal-Flag tagged fragments.

they promote SNARE complex assembly but also because they participate in upstream interactions that provide a bridge between the two membranes.

A function of Munc13s in docking in the traditional sense of bridging two membranes (sometimes referred to as tethering when the intermembrane distances are longer) seems natural given the architecture of their conserved C-terminal region, with an elongated MUN domain that is flanked by domains with demonstrated or putative membrane-binding properties and that is related to tethering factors involved in traffic at diverse compartments (*Li et al., 2011*). Indeed, these tethering factors likely facilitate SNARE complex formation by similar mechanisms to that proposed here for Munc13-1 (*Yu and Hughson, 2010*). The presence of $C_1$ domain and $C_2$ domains adjacent to the MUN domain in Munc13s provides opportunities for modulation by factors that regulate release such as DAG, $PIP_2$ and $Ca^{2+}$. In addition, Munc13-1 contains an N-terminal $C_2A$ domain that contributes to vesicle docking via interaction with αRIMs [(*Dulubova et al., 2005*); M. Camacho and C. Rosenmund, unpublished results], which underlies the finding that rescue of release by $C_1C_2BMUNC_2C$ is incomplete (*Figure 10*) and further supports the notion of an overall role for Munc13s in docking. We also note that reconstitution studies had previously shown that Munc13-4 promotes docking of V- to T-membranes in a $Ca^{2+}$-dependent manner (*Boswell et al., 2012*), but it is unclear to what extent this activity is related to that describe here for Munc13-1 because the C-terminal $C_2$ domain of Munc13-4 binds $Ca^{2+}$, whereas the Munc13-1 $C_1C_2BMUNC_2C$ is not predicted to bind $Ca^{2+}$ (*Rizo and Sudhof, 1998*).

Synaptic vesicle docking and priming occur before $Ca^{2+}$ influx and hence the underlying interactions are not expected to require $Ca^{2+}$. Correspondingly, $C_1C_2BMUNC_2C$ can bridge V- to T-membranes in the absence of $Ca^{2+}$ (*Figure 6*). The interactions that cause such bridging are still unclear and extensive studies will be required to characterize them because the distinct vesicle clustering properties of $C_1C_2BMUN$ and $C_1C_2BMUNC_2C$ (*Figure 2*, *Figure 2—figure supplement 1*, *Figure 6*) suggest that these large protein fragments contain multiple membrane binding sites that can cooperate in cis to interact with a single membrane or in trans to bind to two membranes (*Figure 7—figure supplement 1*). We have been unable to express the $C_2C$ domain alone and we have not detected specific interactions of this domain with the SNAREs in preliminary NMR experiments using the $MUNC_2C$ fragment. Although this fragment does not bind tightly to liposomes (*Figure 6—figure supplement 2*), it seems likely that the $C_2C$ domain contains a lipid-binding site(s) with moderate affinity, as this feature would explain the stronger overall liposome clustering activity of $C_1C_2BMUNC_2C$ compared to $C_1C_2BMUN$ (*Figure 6—figure supplement 1*) and phospholipid binding is a characteristic property of $C_2$ domains (*Rizo and Sudhof, 1998*). We speculate that $Ca^{2+}$-free $C_1C_2BMUNC_2C$ may bridge T- and V-liposomes because the $C_1-C_2B$ region binds to the DAG-$PIP_2$-containing T-liposomes in a configuration that favors binding of the $C_2C$ domain in trans to another membrane, i.e. a V-liposome (*Figure 7*; *Figure 7—figure supplement 1C*). Cooperation between multiple $C_2C$ domains could readily strengthen the V-liposome binding and hence the bridging activity.

$Ca^{2+}$ had generally small effects on vesicle clustering (*Figure 2—figure supplement 1*, *Figure 6C, G*, *Figure 4E*, *Figure 9—figure supplement 3E*) except for a strong stimulation of V-liposome docking by $C_1C_2BMUNC_2C$ (*Figure 6A*) that is unlikely to have physiological relevance. Hence, an effect on docking cannot explain the dramatic effects of $Ca^{2+}$ in the reconstitutions that include $C_1C_2BMUNC_2C$, Munc18-1 and NSF-αSNAP (*Figures 4B,D,9B,D*). Note that the ability of $C_1C_2BMUNC_2C$ to stimulate both lipid and content mixing of V- and T-liposomes is largely independent of $Ca^{2+}$ in the absence of Munc18-1 and NSF-αSNAP (*Figure 5A,B*), as expected because $Ca^{2+}$-free $C_1C_2BMUNC_2C$ clusters V- to T-liposomes (*Figure 6G,H*). In the presence of $C_1C_2BMUNC_2C$, Munc18-1 and NSF-αSNAP, there is substantial lipid mixing but practically no content mixing in the absence of $Ca^{2+}$, and both lipid and content mixing occur very fast upon $Ca^{2+}$ addition (*Figures 4* and *9*; *Figure 9—figure supplement 2*). These observations indicate that partial SNARE complex formation already occurs in the absence of $Ca^{2+}$, resulting in the formation of a 'primed state' that is ready for fusion but only fuses (and fast) upon $Ca^{2+}$ addition [note in this context that the lipid mixing observed before adding $Ca^{2+}$ can occur through lipid transfer when the SNARE complex brings membranes transiently into proximity without the need for fusion or hemifusion (*Zick and Wickner, 2014*)]. Formation of this primed state requires Munc18-1, Munc13-1, NSF, αSNAP and the three SNAREs, but not Syt1 (*Figure 4A,B*). Such requirements correlate with the findings that Munc18-1 and Munc13s are essential for vesicle priming (*Verhage et al., 2000*; *Varoqueaux et al., 2002*) whereas Syt1 is not

(*Geppert et al., 1994*; *Bacaj et al., 2015*), supporting the notion that the primed state formed in our reconstitutions resembles the primed state of synaptic vesicles.

The nature of the primed protein complex underlying this state is unclear, but it is likely to include $C_1C_2BMUNC_2C$ and Munc18-1 since the MUN domain binds to membrane-anchored SNARE complexes (*Guan et al., 2008*) and Munc18-1 also binds to SNARE complexes (*Dulubova et al., 2007*; *Shen et al., 2007*). This proposal is consistent with data suggesting a role for Sec1p (the yeast Munc18-1) after SNARE complex assembly (*Grote et al., 2000*) and with the observation that a constitutively open syntaxin-1 mutant fully rescues the docking defect observed in *Unc13* nulls in C. elegans but rescues neurotransmitter release only partially, which also suggested a role for Unc13 downstream of SNARE complex formation (*Hammarlund et al., 2007*). It is also possible that α-SNAP forms part of this primed complex (*Zick et al., 2015*). Regardless of its composition, our data suggest that the primed state is metastable and requires $Ca^{2+}$ binding to the Munc13-1 $C_2B$ domain for efficient content mixing (*Figure 9F*) either because the $Ca^{2+}$-bound Munc13-1 $C_2B$ domain contributes directly to facilitate membrane fusion or because $Ca^{2+}$ binding releases an inhibitory interaction existing in the primed state. The finding that a mutation in a $Ca^{2+}$-binding loop of the Munc13-2 $C_2B$ domain increases release probability (*Shin et al., 2010*) is consistent with both possibilities and supports the notion that Munc13s form intrinsic part of the primed state of synaptic vesicles.

Clearly, our reconstitutions raise many questions and are incomplete, as they do not incorporate other important proteins that control release such as CAPS, complexins, RIMs or Rab3s (*Rizo and Sudhof, 2012*). While Syt1 $C_2AB$ stimulates content mixing in reconstitutions with $C_1C_2BMUN$ (*Figure 3*), the effects of Syt1 become masked with $C_1C_2BMUNC_2C$ (*Figure 4*), likely because this Munc13-1 fragment promotes fusion with high efficiency and no further acceleration can be observed in our experiments. Thus, effects of Syt1 may only be observable at faster time scales, as Syt1 is not essential for release but is key to trigger release with high speed upon $Ca^{2+}$ influx (*Sudhof, 2013*). Approaches that allow faster measurements [e.g. single vesicle assays (*Lee et al., 2010*; *Kyoung et al., 2011*); *Diao et al., 2012*; *Lai et al., 2014*] will be required to test this prediction. An additional advantage of these assays is that they allow distinction of the docking event from lipid and content mixing. Note also that the strong effect of the D705N,D711N mutation in the Munc13-1 $C_2B$ domain in our reconstitutions (*Figure 9E,F*) contrasts with the mild effects of an analogous mutation in the $C_2B$ domain of the related isoform Munc13-2 on evoked release (*Shin et al., 2010*). However, it is plausible that these mild effects arise because this isoform has some functional differences with Munc13-1 (*Rosenmund et al., 2002*) or because there is some functional redundancy with another protein not included in our reconstitutions (e.g. CAPS), and the increased release probability caused by the mutation in the Munc13-2 $C_2B$ domain $Ca^{2+}$-binding loops (*Shin et al., 2010*) suggests a role as a $Ca^{2+}$ sensor that might cooperate with Syt1. Mutating the Munc13-2 $Ca^{2+}$-binding sites did have marked effects on release during action-potential trains; hence, our reconstitution results, which were obtained in the presence of DAG and $PIP_2$, may be related to hyper-activated states present during these trains.

Further research will be required to distinguish between these possibilities, but we would like to emphasize that our reconstitutions recapitulate many key features of neurotransmitter release. Hence, unexpected findings from our reconstitutions that appear to contradict physiological data may actually uncover novel mechanistic aspects of release that were not observed in physiological experiments because of functional redundancy or compensatory effects by factors not included in the reconstitutions.

## Materials and methods

### Recombinant proteins

Expression and purification of full length rat syxtaxin-1A, full length human SNAP-25A (with its four cysteines mutated to serines), full-length rat Synaptobrevin, full-length rat Munc18-1, the rat synaptotagamin-1 C2AB fragment (residues 131–421), full length Cricetulus griseus NSF V155M mutant, full-length Bos taurus αSNAP, and rat Munc13-1 fragments spanning the MUN and MUNC2C regions (residues 859–1531 and 859–1735, respectively, both with residues 1408–1452 from a

flexible loop replaced by the sequence EF), were described previously (*Ma et al., 2011*; *2013*; *Chen et al., 2002*; *2006*; *Dulubova et al., 1999*; *Xu et al., 2013*).

A construct encoding TM and cytoplasmic regions of Syt1 (residues 57–421 with the following cysteine mutations: C74S, C75A, C77S, C79I, C82L) with a C-terminal His-tag within a Pet28a vector (*Mahal et al., 2002*) was a kind gift from Thomas Sollner. The protein was expressed in Escherichia coli BL21(DE3) cells in Terrific Broth media at 16°C for 18 hr with 0.4 mM isopropyl β-D-1-thiogalacto-pyranoside. Cells were re-suspended in a buffer containing 50 mM Hepes pH 7.4 and 600 mM KCl with a protease inhibitor mixture, and lysed using an Avestin EmulsiFlex-C5 homogenizer. The soluble fraction of the cell lysate was collected after centrifugation at 48,000 × g for 30 min; 1% β-OG was very slowly added to this soluble fraction and incubated on an orbital shaker for 4 hr at 4°C. This mixture was centrifuged at 48,000 × g for 30 min, and the soluble fraction was incubated with Ni-NTA resin (Qiagen; Valencia, CA) at 4°C for 2 hr. The resin was washed with a buffer containing 50 mM Hepes pH 7.4, 600 mM KCl, 10 mM Imidazole and 1% β-OG. Nucleic acid contaminants were cleared with benzonase treatment of the resin using 40 units of benzonase per milliliter of solution. The Syt1 fragment was eluted using the washing buffer supplemented with 250 mM imidazole and further purified with size-exclusion chromatography on a Superdex 200 16/60 column using 20 mM Hepes pH 7.4 containing 600 mM KCl and 1% β-OG as the buffer.

To express rat Munc13-1 fragments encoding the $C_1C_2BMUN$ and $C_1C_2BMUNC_2C$ regions (residues 529–1531 and 529–1735, respectively, both with residues 1408–1452 from a flexible loop replaced by the sequence EF), the corresponding DNA sequences originating from full-length rat Munc13-1 (*Basu et al., 2005*) were cloned into the pFastBac vector (the vector was modified by adding a GST tag and a TEV cleavage site in front of the EcoRI cloning site). The construct was used to generate a baculovirus using the Bac-to-Bac system (Invitrogen). Insect cells (sf9) were infected with the baculovirus, harvested about 72–96 hr post-infection, and re-suspended in lysis buffer (50 mM Tris pH8.0, 250 mM NaCl, 1 mM TCEP). Cells were lysed and centrifuged at 18,000 rpm for 45 min, and the clear supernatant was incubated with GST agarose at room temperature for 2 hr. The beads were washed with: i) lysis buffer; ii) lysis buffer containing 1% TX-100; iii) lysis buffer containing 1M NaCl; and iv) lysis buffer. The protein was treated with TEV protease on the GST agarose at 22°C for 2 hr. The protein was further purified by ion exchange chromatography and gel filtration, and was concentrated to 1–4 mg/ml for storage in 10 mM Tris buffer (pH 8.0) containing 10% glycerol, 5 mM TCEP and 250 mM NaCl. The $C_1C_2BMUNC_2C$ D705N,D711N mutant was generated by site-directed mutagenesis, and purified as the WT fragment.

## Liposome co-floatation assays

Lipids mixture containing 37.5% POPC, 18% POPE, 20% DOPS, 2% PIP$_2$, 2% DAG, 20% Cholesterol and 0.5% Rhodamine-PE were dried in glass tubes with nitrogen gas and kept under vacuum overnight. Lipid films were re-suspended in buffer (25 mM HEPES, pH 7.4, 150 mM KCl, 10% glycerol (v/v)) and vortexed for 5 min. The re-suspended lipid films were frozen and thawed for five times, then extruded through a 80 nm polycarbonate filter with an Avanti extruder for at least 19 times and the size of the liposomes was analyzed by DLS. Liposome solutions containing 2 mM lipids were incubated with 1 μM Munc13-1 fragments at room temperature for 1 hr. The liposomes and bound proteins were isolated by a co-floatation assay on a Histodenz density gradient (40%:35%:30%) as described previously (*Guan et al., 2008*). Samples from the top of the gradient were taken and analyzed by SDS-PAGE and Coomassie blue staining.

## Lipid mixing assays monitored by NBD-fluorescence de-quenching

These lipid mixing assays were performed as described in (*Ma et al., 2013*) with some modifications. Donor liposomes with synaptobrevin (V) contained 40% POPC, 20% DOPS, 17% POPE, 20% Cholesterol, 1.5% NBD PE, and 1.5% Liss Rhod PE. Donor liposomes with both synaptobrevin and Synaptotagmin-1 (VSyt1) contained 40% POPC, 6.8% DOPS, 30.2% POPE, 20% Cholesterol, 1.5% NBD PE, and 1.5% Liss Rhod PE. Acceptor liposomes with syntaxin-1-SNAP-25 (T) contained 38% POPC, 18% DOPS, 20% POPE, 20% Cholesterol, 2% PIP2 and 2% DAG. Lipid mixtures were dried in glass tubes with nitrogen gas and kept under vacuum overnight. Lipid films were re-suspended and dissolved in buffer (25 mM HEPES, pH 7.4, 150 mM KCl, 0.5 mM TCEP, 10% glycerol (v/v)) with 1% β-OG. Purified SNARE proteins containing 1% β-OG were added to liposomes to make Syx:SNAP25:Lipid

ratios = 1:5:800 for T-liposomes, Syb:Lipid = 1:500 for V-liposomes and Syt1:Syb:Lipid = 1:2:1000 for VSyt1 liposome. The mixtures were incubated at room temperature for 30 min and dialyzed against the reaction buffer (25 mM HEPES, pH 7.4, 150 mM KCl, 0.5 mM TCEP, 10% glycerol (v/v)) with 1 g/L Biobeads SM2 (Bio-Rad; Hercules, CA) 3 times.

For lipid mixing assays, donor liposomes (0.125 mM lipids) were mixed with acceptor liposomes (0.25 mM lipids) with various additions of the other proteins in a total volume of 200 µl. For experiments with NSF-αSNAP, acceptor liposomes were first incubated with 0.8 µM NSF, 2 µM αSNAP, 2.5 mM $MgCl_2$, 2 mM ATP, 0.1 mM EGTA and 1 µM Munc18-1 at 37°C for 25 min, and then mixed with donor liposomes and 0.5 µM Munc-13 fragments, 1 µM C2AB fragment, 1 µM excess SNAP-25, and 0.1 mM EGTA (this amount of EGTA is critical to prevent effects from residual $Ca^{2+}$ co-purified with the Munc13-1 fragments and is low enough to preserve the $Zn^{2+}$-binding sites of the $C_1$ domain). To measure the effects of $Ca^{2+}$, 0.6 mM $Ca^{2+}$ was added after 300 s of the start of the reaction. The NBD-PE fluorescence probe was excited at 460 nm and the emission signal from NBD was monitored at 538 nm with a PTI Spectrofluorometer (Edison, NJ). All experiments were performed at 37°C. At the end of each reaction, 1% w/v β-OG was added to solubilize the liposomes, and all the data were normalized to the maximum fluorescence signal achieved after addition of β-OG. All the experiments were repeated at least three times with a given preparation and the results were verified in multiple experiments performed with different preparations. For quantification, we calculated the average fluorescence at 500 s, expressed as percentage of the maximum fluorescence, and the corresponding standard deviation.

## Simultaneous lipid mixing and content mixing assays

Assays that simultaneously measured lipid mixing from de-quenching of the fluorescence of Marina Blue-labeled lipids and content mixing from the development of FRET between PhycoE-Biotin trapped in the T-liposomes and Cy5-Streptavidin trapped in the V-liposomes were performed as described (*Zucchi and Zick, 2011*; *Zick and Wickner, 2014*) with some modifications. V-liposomes with synaptobrevin contained 39% POPC, 19% DOPS, 19% POPE, 20% Cholesterol, 1.5% NBD PE, and 1.5% Marine Blue PE. VSyt1-liposomes with both synaptobrevin and Synaptotagmin-1 contained 40% POPC, 6.8% DOPS, 30.2% POPE, 20% Cholesterol, 1.5% NBD PE, and 1.5% Marine Blue PE. T-liposomes with syntaxin-1-SNAP-25 contained 38% POPC, 18% DOPS, 20% POPE, 20% Cholesterol, 2% $PIP_2$ and 2% DAG. Lipid mixtures were dried in glass tubes with nitrogen gas and under vacuum overnight. Lipid films were re-suspended and hydrated in buffer (25 mM HEPES, pH 7.4, 150 mM KCl, 0.5 mM TCEP, 10% glycerol (v/v)) with 1% β-OG by vortex and sonication. Purified SNARE proteins and fluorescence labeled proteins were added to lipid mixtures to make Syx:SNAP25:Lipid = 1:5:800 and PhycoE-Biotin 4 µM for T-liposomes; Syb:Lipids = 1:500 and Cy5-Streptavidin 8 µM for V-liposomea; and Syt1:Syb:Lipids = 1:2:1000 and Cy5-Streptavidin 8 µM for VSyt1 liposomes. The mixtures were incubated at room temperature for 30 min and dialyzed against the reaction buffer (25 mM HEPES, pH 7.4, 150 mM KCl, 0.5 mM TCEP, 10% glycerol (v/v)) with 1g/L Biobeads SM2 (Bio-Rad) 3 times at 4°C. The proteoliposomes were purified by floatation on a three-layer histodenz gradient (35%, 25%, and 0%) and harvesting from the topmost interface.

To simultaneously measure lipid mixing and content mixing, T-liposomes (0.25 mM lipids) were mixed with V-liposomes (0.125 mM lipids) with various additions in a total volume of 200 µl under analogous conditions as those described above for lipid mixing assays, including 100 µM EGTA. All experiments were performed at 30°C, and 0.6 mM $Ca^{2+}$ was added at 300 s. The fluorescence signals from Marine Blue (excitation at 370 nm, emission at 465 nm) and Cy5 (excitation at 565 nm, emission at 670 nm) were recorded to monitor lipid and content mixing, respectively. At the end of each reaction, 1% w/v β-OG was added to solubilize the liposomes, and the lipid mixing data were normalized to the maximum fluorescence signal achieved after addition of β-OG. To measure content mixing without interference from vesicle leakiness, most experiments were performed in the presence of 5 µM streptavidin. Control experiments without streptavidin were performed to measure the maximum Cy5 fluorescence attainable upon detergent addition. However, there was a large variability in the maximum Cy5 fluorescence values observed, perhaps because of binding of the dye to the detergent. Since the average maximum values observed in detergent were similar to the maximum Cy5 fluorescence observed at the end of the most efficient fusion reactions (those including the Munc13-1 $C_1C_2BMUNC_2C$ fragment, e.g. *Figure 9B*, blue trace) and the latter was more

reproducible, in practice we used averaged maximum values of these reactions for normalization. All the experiments were repeated at least three times with a given preparation and the results were verified in multiple experiments performed with different preparations. For quantification, we calculated the average fluorescence at 500 s, expressed as percentage of the maximum fluorescence, and the corresponding standard deviation.

## Measuring the clustering activity of Munc13-1 fragments by DLS

The clustering activity of Munc13-1 fragments was measured by DLS using a DynaPro instrument (Wyatt Technology; Santa Barbara, CA) basically as described (*Arac et al., 2006*). For experiments with liposomes and Munc13-1 fragments, the conditions and liposome compositions were similar to those of the liposomes used for lipid mixing assays unless specifically indicated. Thus, 0.5 µM Munc13-1 fragments were mixed with T-liposomes (0.25 mM lipids) and/or V-liposomes (0.125 mM lipids) and incubated in a buffer containing 25 mM Hepes (pH 7.4), 150 mM KCl, 10% (v/v) glycerol, 0.5 mM TCEP, 2.5 mM $MgCl_2$, 0.1 mM EGTA at 25°C. For titrations with Munc13-1 fragments (*Figure 6—figure supplement 1*), different concentrations of the fragments were mixed with T-liposomes (0.25 mM lipids) and V-liposomes (0.125 mM lipids) and incubated in the same buffer at 30°C. For experiments to monitor particle size in parallel with lipid and content mixing, lipid and content mixing assays were performed at 30°C under the standard conditions described above but with all protein and liposome concentrations diluted 8-fold, and identical samples were analyzed by DLS as a function of time at 30°C.

## Lentiviral constructs

The cDNAs of Munc13-1 full length and C-terminal fragments ($C_1C_2BMUNC_2C$, aa529-1693; $C_1C_2BMUN$, aa529-1508; MUN domain, aa859-1508; and $MUNC_2C$, aa859-1693) were generated from rat Munc13-1 (*Basu et al., 2005*) by PCR amplification. The reverse primer harbors a 3xFLAG sequence (Sigma-Aldrich) to allow expression analysis. The corresponding PCR products were fused to a P2A linker (*Kim et al., 2011*) after a nuclear localized GFP sequence into the lentiviral shuttle vector, which allows a bicistronic expression of NLS-GFP and the Munc13-1-Flag protein/fragment under the control of a human *SYNAPSIN1* promoter. Concentrated lentiviral particles were prepared as described (*Lois et al., 2002*).

## Autaptic hippocampal neuronal cultures and lentiviral infection

Animal welfare committees of Charité Medical University and the Berlin state government Agency for Health and Social Services approved all protocols for animal maintenance and experiments (license no. T 0220/09). Astrocytes were plated at a density of 5000 cells/$cm^2$ onto microdots coated coverslips to allow them to grow onto the growth permissive substrate. Hippocampi were dissected from embryonic day 18.5 Munc13 1/2 DKO mouse and enzymatically treated with 25 units $ml^{-1}$ of papain for 45 min at 37°C. After enzyme digestion, hippocampi were mechanically dissociated and the neuron suspension was plated onto the astrocytes microislands at a final density of 300 cells $cm^{-2}$. Neurons were incubated at 37°C and 5% $CO_2$ for 13–16 days to mature before starting the experiments; 24 hr after plating, neurons were infected with lentiviral rescue constructs per 35 mm diameter well.

## Electrophysiology

Synaptic function was assayed by whole-cell voltage clamp. Synaptic currents were monitored using a Multiclamp 700B amplifier (Axon instrument). The series resistance was compensated by 70% and only cells with series resistances <10 MΩ were analyzed. Data were acquired using Clampex 10 software (Axon instrument) at 10 kHz and filtered using a low-pass Bessel filter at 3 kHz. Recordings were done at room temperature in autaptic hippocampal Munc13- 1/2 DKO neurons at 13–16 days in vitro (DIV). Borosilicate glass pipettes with a resistance between 2–3.5 MΩ were used. Internal pipette recording solution contained the following (in mM): 136 KCl, 17.8 HEPES, 1 EGTA, 4.6 $MgCl_2$, 4 $Na_2ATP$, 0.3 $Na_2GTP$, 12 creatine phosphate, and 50 $Uml^{-1}$ phosphocreatine kinase; 300 mOsm; pH 7.4. Neurons were continuously perfused with standard extracellular solution including the following (in mM): 140 NaCl, 2.4 KCl, 10 HEPES, 10 glucose, 2 $CaCl_2$, 4 $MgCl_2$; 300 mOsm; pH 7.4. Action potential-evoked EPSCs were triggered by 2 ms somatic depolarization from −70 to 0

mV. To determine the size of the readily-releasable pool (RRP), hypertonic solution, 500 mM sucrose added to standard extracellular solution, was applied directly onto the isolated neuron for 5s using a fast application system. A transient inward current component that lasts for 2–3 s represents the RRP charge (*Rosenmund and Stevens, 1996*).

### Western blot

Hippocampal neurons from E18.5 Munc13-1/2 DKO at a density of 10.000 / cm$^2$ were plated into 6 well plates containing monolayer cultures of astrocytes. Neurons were lysed after 15 DIV at 4°C with 50 mM Tris·HCl, pH 7.9, 150 mM NaCl, 5 mM EDTA, 1% Triton X-100, 1% sodium deoxycholate, 250 µM phenylmethylsulfonyl fluoride, 1% Nonidet P-40, and protease inhibitor cocktail-complete mini (Roche Diagnostics, Berlin, Germany). Lysates were mixed with Laemmli Buffer containing 0.3 mM DTT, and boiled 10 min at 95°C. 30 µg of protein lysates were used for the SDS-PAGE electrophoresis. After separation by SDS-PAGE proteins were transferred to a polyvinyl difluoride (PVDF) membrane. Membranes were blocked with 5% skim milk in TBST, and incubated at 4°C *over night* with primary antibodies: anti-Flag M2 (F1804; Sigma-Aldrich), and anti-Living Colors GFP (632375; Clontech; Mountain View, CA). Secondary antibodies were horseradish peroxidase-conjugated (Jackson ImmunoResearch; West Grove, PA). The immunoreactive proteins were detected by ECL Plus Western Blotting Detection Reagents (GE Healthcare Biosciences; Pittsburgh, PA) in a Fusion FX7 detection system (Vilber Lourmat, Eberhardzell, Germany).

## Acknowledgements

We thank Shae Padrick for fruitful discussions and the Charite viral core facility for provision of lentiviral constructs. This research was supported by a grant from the Welch Foundation (I-1304) (to JR), by NIH grants NS037200 and NS040944 (to JR), by German Research Council grant SFB958 (to CR) and by the ERC grant SynVglut (to CR).

## Additional information

### Competing interests

CR: Reviewing editor, *eLife*. The other authors declare that no competing interests exist.

### Funding

| Funder | Grant reference number | Author |
| --- | --- | --- |
| German Research Council | SFB958 | Christian Rosenmund |
| European Research Council | SynVglut | Christian Rosenmund |
| Welch Foundation | I-1304 | Josep Rizo |
| National Institutes of Health | NS037200 | Josep Rizo |
| National Institutes of Health | NS040944 | Josep Rizo |

The funders had no role in study design, data collection and interpretation, or the decision to submit the work for publication.

### Author contributions

XL, ABS, MC, VE, Conception and design, Acquisition of data, Analysis and interpretation of data, Drafting or revising the article; JX, TT, BQ, Acquisition of data, Analysis and interpretation of data; LS, Conception and design; CM, CR, JR, Conception and design, Analysis and interpretation of data, Drafting or revising the article

### Author ORCIDs

Cong Ma, http://orcid.org/0000-0002-7814-0500
Josep Rizo, http://orcid.org/0000-0003-1773-8311

### Ethics

Animal experimentation: Animal welfare committees of Charité Medical University and the Berlin state government Agency for Health and Social Services approved all protocols for animal maintenance and experiments (license no. T 0220/09).

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
