## [Decision Letter]

Thank you for submitting your work entitled "Synergy between Munc13 and Synaptotagmin C_2_ domains in Synaptic Vesicle Fusion" for consideration by *eLife*. Your article has been evaluated by Randy Schekman as Senior Editor and three peer reviewers, one of whom is a member of our Board of Reviewing Editors.

The reviewers have discussed the reviews with one another and the Reviewing Editor has drafted this decision to help you prepare a revised submission. Major revisions are needed before a final decision can be made.

Summary:

In this work the function of Munc13 is studied using (1) reconstituted bulk liposome content and lipid mixing assays and (2) electrophysiology experiments in neuronal cultures with rescue using Munc13 DKO hippocampal neurons. The results suggest that the Munc13 C_2_ domains (especially C_2_B and C_2_C) play key roles in priming as demonstrated by Figure 9 (reconstitution experiments) and 10 (neuronal culture experiments). It is also suggested that the Munc13 C_2_ domains act synergistically together with those of synaptotagmin upon Ca^2+^-triggering. The study extends a previous one by the authors (Ma et al., 2013) in fundamental ways.

Since fusion reactions were monitored in bulk, the effects on fusion efficiency cannot be separated from effects on vesicle docking, posing questions on the proposed role on intrinsic fusion mechanisms (points 2, 3, 6 and 9). Moreover, the absence of complexin is a limitation in their reconstituted experiments and may have influenced the effect of Munc13 in their in vitro experiments. While the experiments presented in this work expand on previous work on the role of Munc13 on synaptic vesicle priming and plasticity, the suggested role as acting in "synergy" with the synaptotagmin C_2_ domains is less clear and, ideally, should be further tested by additional experiments as outlined below (points 1, 2, and 3). The role of Munc13 in fusion per se is interesting, but the available data are still a bit too inconclusive to make such a strong statement – even in the title. For example, the DD/NN mutation in C_2_B impairs liposome fusion, but corresponding mutations do not affect evoked EPSCs. It might be prudent to discuss these data a bit more carefully.

The manuscript is rather cumbersome to understand and there are rather different results obtained with the soluble Syt2 C_2_AB domain vs. full-length Syt1. Thus the work would benefit from elimination of all results with the soluble Syt1 C_2_ AB domain (points 5, 8, 9), also considering that full-length Syt1 is more biologically relevant.

Essential revisions:

1) Figure 9 suggests that the Ca^2+^ binding region of the Munc13 C_2_B is important for enhancing Ca^2+^ triggered content/lipid mixing in a bulk assay with SNAREs, full-length Syt1, αSNAP, NSF, and C_1_ C_2_BMUNC_2_C (the figure shows that the DN mutant of the Munc13 C_2_B domain has a reduced Ca^2+^ triggered lipid/content mixing effect). This result is at odds with the results by Shin et al. (2010) using rescue experiments with Munc13-2 in cultured hippocampal neuronal cultures derived from Munc13-1/2 DKO mice. The experiments of Shin et al. revealed no effect of the DN mutant on isolated EPSCs, and on the RRP. However, Shin et al. observed an effect on short term plasticity that was explained by an induced increase in PIP2 concentration upon high frequency stimulation. This discrepancy between Figure 9 and the EPSC/RRP results by Shin et al. needs to be discussed, and follow up experiments may be necessary (see next point).

2) As is apparent from both the lipid and content mixing traces in Figure 9, spontaneous fusion reactions occur at a substantial rate before Ca^2+^ is injected, i.e., the system has not reached a steady state level at the time point of Ca^2+^ injection. Since in their bulk assay, t- and v-vesicles may continuously undergo docking, this produces opportunities for both lipid and content mixing to occur as a consequence of an increasing number of docked vesicles. Upon injection of Ca^2+^, the observed burst of both content and lipid mixing signals could be a consequence of an increased probably of vesicle docking upon Ca^2+^ binding to Munc13-C_2_B rather than an increase of intrinsic fusion probability of already docked vesicle pairs. Thus, it would important to perform clustering experiments plus/minus Ca^2+^ with wildtype C_1_ C_2_BMUNC_2_C and the DN mutant under the same conditions as Figure 9 in order to assess the potential effect of the Ca^2+^ bound C_2_B domain of Munc13 on vesicle clustering.

3) Figure 6 suggests that C_1_ C_2_BMUNC_2_C greatly enhances both content and lipid mixing in the absence of Ca^2+^ together with SNAREs, compared to just SNAREs alone. However, this result could also be a consequence of enhanced docking by C_1_ C_2_BMUNC_2_C in the absence of Ca^2+^, rather than an effect on intrinsic spontaneous fusion itself. To assess this question, clustering experiments should be attempted with exactly the same conditions and constructs to assess if C_1_C_2_MUNC_2_C enhances vesicle clustering.

4) Figure 7 suggests that injection of Ca^2+^ does not induce any increase in lipid/content mixing in presence of SNAREs and full length Syt1 (orange curve, i.e., no Munc18, Munc13, αSNAP/NSF). This is at odds with previous reconstitution experiments that showed an increase of both properties with SNAREs and Syt1 alone upon Ca^2+^ injection (Malsam et al., EMBO J 2012; Lai et al., *eLife* 2014). An explanation is needed.

5) The effect of Munc13 in spontaneous (Ca^2+^ independent) and Ca^2+^-triggered fusion are very different when Syt1-C_2_AB fragment or reconstituted full-length Syt1 is used (compare Figure 5 and Figure 7), indicating that Syt1-C_2_AB behaves quite differently from full-length Syt1 in their reconstituted system. The use of reconstituted full-length Syt1 is more physiological. For the benefit of a clearer presentation of the results, all experiments with Syt1 C_2_AB could be eliminated from the paper.

6) Figure 6 and Figure 7 show the effect of C_1_C_2_BMUN on lipid mixing in the absence or presence of NSF/SNAP and Munc18-1, by the different effects of calcium in this context. The authors suggest that different mechanisms are at play, but there may be other possibilities. The calcium independent lipid mixing effects of C_1_C_2_BMUN are thought to reflect support of SNARE complex assembly by C_1_C_2_BMUN. In the presence of NSF/SNAP and Munc18-1 there are at least two explanations: (1) NSF/SNAP reverts the C_1_C_2_BMUN-mediated SNARE complex assembly in the absence of calcium, while in the presence of calcium more C_1_C_2_BMUN is recruited to the liposomes and can then override the antagonistic effect of NSF/SNAP. (2) The membrane recruited C_1_C_2_BMUN even protects assembled SNARE complexes from NSF/SNAP.

7) Surprisingly, Syt1 does not appear to be essential to control either the speed or the calcium-sensitivity of the reaction when the longest forms of Munc13 are included in their assay (Figure 5). Thus, in this simplified system, the calcium-switch is encoded entirely within the C_2_ domains of Munc13 at a calcium sensitivity somewhere below 100 μM. Establishing this limit with some precision would help ascertain how the activity of this system compares to the resting state in the synapse. The authors also speculate that Syt1-dependence may become apparent if they had faster time resolution in their assay. Given the central role of this calcium-sensor in synaptic biology, establishing whether it has any activity here is essential. However, as pointed out in (5) above, the results with the soluble Syt1 C_2_AB domain are rather different from the results with full-length Syt1. Thus, it might be prudent to eliminate all results with the Syt1 C_2_AB domain and focus on full-length reconstituted Syt1 throughout this work.

8) When shorter versions of Munc13 are tested where the C_2_C domain is missing, the authors now see Syt1-dependent effects on membrane fusion. In particular, they make the important point that lipid-mixing proceeds "efficiently" without Syt1, while contents-mixing is strongly enhanced by Syt1 C_2_AB. However, in Figure 4, it is not obvious that the effect of Syt1 is more pronounced for content vs lipid-mixing. Although the change in the curves for content-mixing appear more dramatic than for lipid-mixing, this seems to be due to the scale of the axes and not necessarily to the rates. The initial rate changes about 3 fold for contents mixing (comparing B and D). A similar increase in lipid mixing would be hard to see at their resolution, but looks plausible when comparing A and C. A similar burst in lipid-mixing appears to be present in Figure 5. Without meaningful resolution over this range, it is challenging to draw the conclusion "the finding that Syt1 C_2_AB selectively enhances content mixing but not lipid mixing provides strong evidence that Syt1 plays a direct role in membrane fusion…". The same point is echoed strongly in the Discussion. At a minimum, it would be helpful if the authors provided initial rates from each experiment, rather than endpoints. Otherwise, the discussion surrounding whether Syt1 is uniquely designed to support full-fusion should probably be more guarded. However, all these results were generated with the soluble Syt1 C_2_AB, so again, it might be better to eliminate all results with the Syt1 C_2_AB domain from this work.

9) In the Introduction, the authors write "…the mechanism by which Munc13s play an additional role in vesicle docking… is unclear." The papers cited here do actually propose a mechanism for the role of vesicle docking by Munc13s, i.e. docking/membrane attachment, priming, and SNARE complex assembly may be manifestations of the same process, that is, that Munc13s mediate vesicle priming by promoting SNARE complex assembly (i.e., by opening the Munc18-syntaxin complex and catalyzing trans SNARE complex formation). The data provided in these papers are quite compelling. The three papers should be cited separately, including the corresponding conclusions, and then explain why the authors think that these papers are not helpful or believable in defining (part of) the mechanism by which Munc13 mediates vesicle docking. Somewhat in the same context, can one use the liposome-clustering activity of proteins and protein fragments as a basis to extrapolate towards corresponding roles of the respective proteins in synaptic vesicle docking in neurons? For example, in the case of Syt1, such a conclusion can probably not be drawn – loss of Syt1 does not really affect synaptic vesicle docking much.

[Editors' note: further revisions were requested prior to acceptance, as described below.]

Thank you for resubmitting your work entitled "Functional Synergy between the Munc13 C-terminal C_1_ and C_2_ domains" for further consideration at *eLife*. Your revised article has been favorably evaluated by Randy Schekman (Senior editor) and three reviewers, one of whom is a member of our Board of Reviewing Editors.

The manuscript has been improved but there are some relatively minor remaining issues that need to be addressed before acceptance, as outlined below:

The reviewers would like to thank the authors for addressing the concerns raised previously. Based on their new findings, the authors have changed the conclusions from their studies. Rather than suggesting a synergism between the synaptotagmin and Munc13 C_2_ domains, they now suggest synergism between the C_2_B and C_2_C domains of Munc13-1 and that Munc13-1 may bridge V- and T-liposomes in a calcium independent manner (Figure 6—figure supplement 5). The authors' new data correlating the activation of SNARE fusion with (at least in part) a role in the specific clustering/docking of V- and T-liposomes by Munc13 significantly clarifies the primary message of the paper and is a result that will be of general interest.

Comments:

1) The manuscript remains very long. Considering the revised conclusions, the authors are encouraged to take a fresh look and examine if all the data are really essential for the key conclusions.

2) There are some supplementary figures that are actually quite important (e.g., Figure 4—figure supplement 4, and Figure 6—figure supplement 5). Please restrict supplementary figures to repeat experiments, raw data, and non-essential experiments.

3) A general comment regarding all figures that compare lipid and content mixing results (Figure 4, Figure 5, Figure 7, Figure 8, Figure 9 and numerous supplemental figures): both types of experiments (lipid mixing and content mixing, respectively) are normalized differently: for lipid mixing 1% BOG is added and the resulting fluorescence intensity value is used for normalization; whereas for content mixing, it is the maximum value for each group of conditions in the particular panel. So, 100% content mixing could actually correspond to substantially fewer vesicles undergoing content mixing than vesicles undergoing lipid mixing. A suggestion would be to change all figure panel titles from "lipid mixing" to "relative lipid mixing" and "content mixing" to "relative content mixing", and to change all the y-axis labels from "Fluorescence (% of max)" to "Fluorescence (arbitrary units)". Another comment: ensemble experiments cannot determine what fraction of vesicles that undergo lipid mixing also undergo content mixing.

4) Figure 4—figure supplement 4. We appreciate that the authors have performed DLS measurements to address the concern that the large increase in content mixing upon calcium addition could be due to a calcium-dependent docking/clustering effect. However the presence of large vesicles might dominate in the intensity autocorrelation functions shown. Percent Intensity plots (similar to those shown in Figure 2 and Figure 6) should also be provided for Figure 4—figure supplement 4.

5) Figure 4—figure supplement 4 was only done for the soluble Syt1 C_2_AB fragment. If at all possible, please also perform the DLS experiments also with full-length Syt1. The reason is that there is much less lipid mixing in these experiments prior to calcium addition than for the experiments with the soluble Syt1 C_2_AB fragment (compare Figure 7 and Figure 8).

6) Figure 9: It is perhaps possible that there could be compensating factors in neurons that do not show a large phenotype for the DN mutant of Munc13. Nevertheless, if the author's model of calcium-independent bridging of T and V-vesicles by C_1_C_2_BMUNC_2_C is correct (Figure 6—figure supplement 5), why would calcium binding to the C_2_B domain of Munc13 have such a large effect? Please discuss.

7) Figure 6—figure supplement 5: While the preference for the C_1_C_2_B domain for the T-vesicle membrane can be explained by the presence of DAG and PIP2 in that membrane, what is the mechanism that C_2_C would only interact with the V-vesicle membrane?

8) Figure 2 and Figure 6: why does C_1_C_2_BMUN cluster V-vesicles, whereas C_1_C_2_BMUNC_2_C does not? This question is also related to the Discussion section, paragraph five. Please provide an explanation why the C_2_C domain would favor trans-interactions with V-vesicles.

9) Subsection “Simultaneous evaluation of lipid and content mixing”, paragraph two": "A problem…". Actually, synaptic vesicles are acidic, so the property of sulforhodamine to produce an acidic interior of the proteoliposomes is not necessarily a problem. Moreover, the lack of leakiness under such conditions is desirable. In any case, it is good that the authors confirmed the content mixing results with the different content mixing assay by Zucchi and Zick. Thus, there is really no "problem" with the sulforhodamine content mixing assay.

10) Figure 1: the effect of the different lipid compositions (PIP2 and DAG) appears to be much more pronounced in the presence of NSF/SNAP/Munc18, although C_1_C_2_BMUN is present in both experiments.

11) Discussion, paragraph seven: in addition to Lee and Kyoung, please also cite the improvements of this assay by Diao et al. (*eLife* 2012); Lai et al. (*eLife* 2014). Another comment: the single vesicle content mixing assay by Kyoung et al. (2011); Lai et al. (2014) allows discrimination between effects that are due to docking vs. due to fusion probabilities in addition to the improved time resolution offered by such single vesicle assays.

[Editors' note: further revisions were requested prior to acceptance, as described below.]

Thank you for resubmitting your work entitled "Functional Synergy between the Munc13 C-terminal C_1_ and C_2_ domains" for further consideration at *eLife*. Your revised article has been favorably evaluated by Randy Schekman (Senior editor) and a Reviewing Editor.

The manuscript has been improved but there are some remaining issues that need to be addressed before acceptance, as outlined below:

We thank the authors for their thoughtful considerations of the concerns raised previously. However, the Reviewing Editor has two remaining concerns:

1) Related to Point 9, the authors write:

"A potential concern with the use of sulforhodamine B fluorescence de-quenching to monitor content mixing in these bulk reconstitution experiments is that de-quenching can also arise from liposome leakiness. We attempted to perform experiments where both liposome populations are loaded with sulforhodamine B to assess how much of the de-quenching arises from leakiness (Yu et al., 2013). However, we were unable to observe lipid mixing in these control experiments. Since the solutions of high sulforhodamine B concentrations trapped in the liposomes are highly acidic (pH about 2), which may underlie the lack of lipid mixing, we attempted to increase the pH of these solutions to 5.5 or 7.4 before trapping them into the liposomes, but at both pH values the liposomes were strongly leaky."

Their observation of lack of lipid mixing when both vesicle populations are filled with sulforhodamine B is strange, considering that the authors observed efficient lipid mixing when only one class of vesicles is filled (Figure 3—figure supplement 1). Moreover, Kyoung et al., 2011 and Diao et al. 2012 observed both lipid and content mixing using their sulforhodamine B content mixing method, with only an extremely small fraction of leakage (0.01% of the docked vesicles). Could the problems be related to differences in reconstitution methods, rather than an issue with the sulforhodamine method per se?

This Reviewing Editor also would like to point out that the pH of the interior of the synaptobrevin/synaptotagmin vesicles ("donor" vesicles) is likely more neutral rather than pH 2 when using the published reconstitution protocol (Kyoung et al. Nat Protoc 8(1):1-16, 2013): the initial protein/lipid/detergent solution uses a detergent concentration that is at the CMC (OG ~ 30 mM), so vesicles do not yet form at this stage. This solution is then injected on a Cl^-^4B column that has been pre-equilibrated with vesicle buffer (pH 7.4, without sulforhodamine B), followed by elution with vesicle buffer (pH 7.4, again without sulforhodamine B) (Kyoung et al. Nat Protoc 8(1):1-16, 2013). Vesicles are forming on the column in a time course of several minutes as the detergent concentration decreases. One would expect that the pH of the interior of the column rapidly reaches 7.4.

In any case, as the authors point out, the alternative PhycoE-Biotin/Cy5-Streptavidin content mixing assay produces similar results (Figure 3—figure supplement 3). However, as written, the second paragraph of subsection “Simultaneous evaluation of lipid and content mixing” may come across to the casual reader as if there is something wrong with the sulforhodamine B content mixing method. Since this is a rather technical issue that distracts from the main messages of this paper, and that may be related to differences in reconstitution methods, the authors may wish to consider deleting this entire paragraph. In the future, perhaps the authors may wish to investigate a variety of content mixing assay jointly with other groups who have developed these assays for a more technical publication.

2) Point 11, related to the sentence starting "Approaches that allow faster measurements [e.g. single vesicle assays (Lee et al., 2010; Kyoung et al., 2011); will be required to test this prediction. " It is not just the faster time resolution that is important for future studies. The single vesicle-vesicle lipid/content mixing assay by Kyoung et al. 2011, Diao et al., 2012 et al. (but not the assay by Lee et al., 2010) can distinguish between effects due to docking, lipid mixing and content mixing on a 100 msec time scale for each individual vesicle pair. In addition to the improved time resolution, the discrimination between effects due to docking and/or fusion will be important for future studies. The authors may wish to consider making the discussion of the relevant publications clearer.

---

## [Author Response]

*Summary: In this work the function of Munc13 is studied using (1) reconstituted bulk liposome content and lipid mixing assays and (2) electrophysiology experiments in neuronal cultures with rescue using Munc13 DKO hippocampal neurons. The results suggest that the Munc13 C2 domains (especially C2B and C2C) play key roles in priming as demonstrated by Figure 9 (reconstitution experiments) and 10 (neuronal culture experiments). It is also suggested that the Munc13 C2 domains act synergistically together with those of synaptotagmin upon Ca2+-triggering. The study extends a previous one by the authors (Ma et al., 2013) in fundamental ways. Since fusion reactions were monitored in bulk, the effects on fusion efficiency cannot be separated from effects on vesicle docking, posing questions on the proposed role on intrinsic fusion mechanisms (points 2, 3, 6 and 9). Moreover, the absence of complexin is a limitation in their reconstituted experiments and may have influenced the effect of Munc13 in their in vitro experiments. While the experiments presented in this work expand on previous work on the role of Munc13 on synaptic vesicle priming and plasticity, the suggested role as acting in "synergy" with the synaptotagmin C2 domains is less clear and, ideally, should be further tested by additional experiments as outlined below (points 1, 2, and 3). The role of Munc13 in fusion per se is interesting, but the available data are still a bit too inconclusive to make such a strong statement – even in the title. For example, the DD/NN mutation in C2B impairs liposome fusion, but corresponding mutations do not affect evoked EPSCs. It might be prudent to discuss these data a bit more carefully.*

The manuscript is rather cumbersome to understand and there are rather different results obtained with the soluble Syt2 C2AB domain vs. full-length Syt1. Thus the work would benefit from elimination of all results with the soluble Syt1 C2AB domain (points 5, 8, 9), also considering that full-length Syt1 is more biologically relevant.

Please note that we have included a new Figure 6 and several new supplementary figures, and we have renumbered some of the figures: the previous Figure 3 is now Figure 3—figure supplement 1; and the previous Figure 4, Figure 5 and Figure 6 are now Figure 3, Figure 4 and Figure 5, respectively.

A) We appreciate the assessment that our paper extends the previous Ma et al. 2013 paper in fundamental ways. We also feel indebted to the reviewers because their thoughtful and detailed comments have helped to make the paper of higher quality and more fundamental (see in particular point B below).

B) We agree that it was premature to draw the strong conclusions on a role of the Munc13-1 C_2_ domains in fusion that we had described in the original manuscript, and we found particularly valuable the concerns from the reviewers regarding the possibility that at least some of the effects observed in the fusion assays arise from effects on clustering. While in the original manuscript we did conclude that the tendency of the Munc13-1 fragments to cluster vesicles contribute to their activity in the fusion assays, we did not make this a major point of the paper because the clustering data did not appear to explain the much stronger stimulation of fusion by C_1_C_2_BMUNC_2_C compared to C_1_C_2_BMUN nor the dramatic stimulation of fusion by Ca^2+^ in the reconstitutions that included Munc18-1, NSF-αSNAP and Munc13-1 fragments. We had performed a detailed analysis of the factors that determine the clustering ability of C_1_C_2_BMUN (shown in Figure 2) and had not seen any strong effects of Ca^2+^ on clustering (shown in new Figure 2—figure supplement 1). We also felt that it was unlikely that the strong effects of the C_2_C domain in the fusion assays could arise from the slightly higher tendency of C_1_C_2_BMUNC_2_C to cluster mixtures of V- and T-liposomes, compared to C_1_C_2_BMUN (previous Figure 2—figure supplement 1 that is now replaced with new data that are shown in Figure 6—figure supplement 1 and was performed under conditions analogous to those of the fusion assays).

We thought that the C_2_C domain was simply adding an additional lipid-binding site to the C_1_C_2_ BMUN sequence but there was an important property that we had not anticipated. Because of the overall concern of the reviewers on the clustering issue, we investigated the clustering properties of C_1_C_2_BMUNC_2_C in more detail and found that C_1_C_2_BMUNC_2_C was unable to cluster V-liposomes in the absence of Ca^2+^ (new Figure 6 and new Figure 6—figure supplement 3), in contrast to C_1_C_2_BMUN (Figure 2). Importantly however, C_1_C_2_BMUNC_2_C does cluster all liposomes in mixtures of V-liposomes and T-liposomes (new Figure 6), showing that C_1_C_2_BMUNC_2_C favors bridging of V-liposomes to T-liposomes. These results and the overall data presented in the paper have led us to emphasize in the revised manuscript the notion that Munc13-1 may bridge synaptic vesicles to the plasma membrane to contribute to docking and facilitate SNARE complex formation. We believe that this notion makes the paper stronger and with a clearer message that makes a lot of sense because the highly conserved C-terminal region of Munc13s is formed by an elongated MUN domain related to tethering factors that function in diverse membrane compartments, and by adjacent C_1_ and C_2_ domains with demonstrated or potential membrane-binding properties.

Because of these new data, and in response to the additional reviewer concerns described in the summary above, in the revised paper we have changed the title and we have drastically toned-down conclusions about a role for the Munc13 C_2_ domains in membrane fusion and about synergy with the synaptotagmin C_2_ domains. However, because it seems unlikely that clustering alone can explain the strong stimulation of fusion by Ca^2+^ (see point 2), we still mention that our data suggest the formation of a primed state that needs Ca^2+^ to trigger fast fusion and we make the suggestion that Munc13 may function in docking, priming and fusion. We hope that the reviewers will agree that our overall data support this suggestion even though do not fully prove it.

C) With regard to some of the other general concerns, we are well aware that our reconstitutions have their limitations, which we point out in the Discussion. However, we would like to emphasize that reconstitution experiments are not physiological by definition. The hope is that by including enough components we can get results that recapitulate at least to a certain extent what happens in vivo. Arguments commonly used to reach such conclusion include: i) there are clear correlations between the reconstitution data and physiological results; and ii) the biochemical properties underlying the reconstitution results make overall sense in the context of what is known about this system from research by a variety of approaches. Thus, many of the papers in the field that used reconstitutions have provided important insights even though they had striking contradictions with physiological data, most notably the fact that lipid and/or content mixing did not require Munc18-1 and Munc13s. Our reconstitutions are the only ones so far that can account for the dramatic disruption of release observed in the absence of Munc18-1 or Munc13s in vivo (Figure 8 is particularly striking in this regard), they include the (arguably) eight most central components of the release machinery, and there are many correlations between the results obtained with these reconstitutions and physiological data (see Discussion).

For these reasons, we believe that our reconstitutions provide a great framework to try to understand the mechanism of release and that biochemical properties that explain our data (e.g. the V- to T-liposome clustering activity of C_1_C_2_BMUNC_2_C) are likely to underlie at least in part the mechanism of action of the same proteins in vivo. Similarly, if some of our reconstitution data appear to contradict physiological results (e.g. the effects of the DD/NN mutation in the Munc13-1 C_2_B domain), it is probable that there is an explanation due to functional redundancy or compensatory effects in vivo. In any case, we agree with the reviewers that we should be prudent when drawing conclusions because it is very difficult to have definitive proof for molecular mechanisms. Hence, we have toned down any strong conclusions we made in the manuscript and tried to use soft terms such as ‘suggest’ or ‘indicate’. We have also tried to improve the discussion of the results obtained with the DD/NN mutation but without extending it to avoid further lengthening the manuscript.

D) With regard to the absence of complexin in our experiments, this protein is believed to inhibit release in the absence of Ca^2+^ and to stimulate release upon Ca^2+^ influx. Hence, it would not be expected to have much effect in our optimal reconstitutions because there is little fusion before adding Ca^2+^ and fast fusion upon Ca^2+^ addition in the time scale of our experiments (e.g. Figure 8). Indeed, in unpublished experiments we have found that complexin-1 has no measurable effects in our fusion assays, but we prefer not to include these data because we believe that the roles of complexin will be better studied with methods that have faster time scales than our bulk assays (as has already been shown in reconstitution studies of complexin function by the labs of Axel Brunger and Yeon-Kyun Shin). At the end of the Discussion we do point out that our reconstitutions are incomplete and miss several important factors (not only complexin).

E) We have tried to improve the presentation of the results, adding substantial new data without lengthening the manuscript too much. We are reluctant to remove the results obtained with the soluble Syt1 C_2_AB fragment because these results are important for how the story is developed (e.g. the switch to a different method of monitoring lipid and content mixing) and because, based on the abundant data already available in the literature with this fragment, it is natural that many researchers in the field would wonder what happens if we add this fragment. Note that experiments using full-length Syt1 do have the advantage of being more physiological, as pointed out in the review, but do not allow a direct comparison of the results with and without the Syt1 C_2_ domains with the same liposome preparations. For instance, there is a clear increase in content mixing when Syt1 C_2_AB is added in reconstitutions with C_1_C_2_BMUN (Figure 4), but it would be more difficult to reach this conclusion comparing results obtained with different V-liposome preparations that include or lack full-length Syt1. This result is important because it shows an effect of the Syt1 C_2_ domains in this system and it is likely that this effect cannot be observed when we use C_1_C_2_BMUNC_2_C because the system is too efficient (see Discussion). Finally, we would like to emphasize that, while it is true that some results obtained with C_2_AB and full-length Syt1 are different (mostly in the absence of Ca^2+^, Munc18-1 and NSF-αSNAP) the data obtained in the presence of Munc18-1, NSF-αSNAP and different Munc13-1 fragments are quite similar for C_2_AB and full-length Syt1 (compare Figure 4 with Figure 8).

Essential revisions:

1) Figure 9 suggests that the Ca^2+^ binding region of the Munc13 C_2_B is important for enhancing Ca^2+^ triggered content/lipid mixing in a bulk assay with SNAREs, full-length Syt1, αSNAP, NSF, and C_1_C_2_BMUNC_2_C (the figure shows that the DN mutant of the Munc13 C_2_B domain has a reduced Ca^2+^ triggered lipid/content mixing effect). This result is at odds with the results by Shin et al. (2010) using rescue experiments with Munc13-2 in cultured hippocampal neuronal cultures derived from Munc13-1/2 DKO mice. The experiments of Shin et al. revealed no effect of the DN mutant on isolated EPSCs, and on the RRP. However, Shin et al. observed an effect on short term plasticity that was explained by an induced increase in PIP2 concentration upon high frequency stimulation. This discrepancy between Figure 9 and the EPSC/RRP results by Shin et al. needs to be discussed, and follow up experiments may be necessary (see next point).

This discrepancy was discussed in the original manuscript. We have tried to improve the discussion of this issue, as mentioned in point C of our response to the Summary.

2) As is apparent from both the lipid and content mixing traces in Figure 9, spontaneous fusion reactions occur at a substantial rate before Ca^2+^ is injected, i.e., the system has not reached a steady state level at the time point of Ca^2+^ injection. Since in their bulk assay, t- and v-vesicles may continuously undergo docking, this produces opportunities for both lipid and content mixing to occur as a consequence of an increasing number of docked vesicles. Upon injection of Ca^2+^, the observed burst of both content and lipid mixing signals could be a consequence of an increased probably of vesicle docking upon Ca^2+^ binding to Munc13-C_2_B rather than an increase of intrinsic fusion probability of already docked vesicle pairs. Thus, it would important to perform clustering experiments plus/minus Ca^2+^ with wildtype C_1_C_2_BMUNC_2_C and the DN mutant under the same conditions as Figure 9 in order to assess the potential effect of the Ca^2+^ bound C_2_B domain of Munc13 on vesicle clustering.

We have performed the requested clustering assays but with reactions diluted 8-fold to prevent saturation of the DLS detector (Figure 9—figure supplement 2 for experiments with VSyt1 vesicles; Figure 4—figure supplement 4 for experiments with V-vesicles and not Syt1). The immediate effects of Ca^2+^ addition on the DLS data are minimal and correlate with the observation that, in general, Ca^2+^ does not have strong effects on clustering ability (except for clustering of V-vesicles by C_1_C_2_BMUNC_2_C, which is unlikely to have any physiological relevance). Hence, it is unlikely that an effect on clustering underlies the drastic stimulation of content mixing induced by Ca^2+^.

We would like to clarify that there is substantial lipid mixing before Ca^2+^ addition, but very little content mixing. In Figure 9—figure supplement 1, we now present quantification of lipid mixing at 300 s, before adding Ca^2+^, in addition to the quantification at 500 s (200 s after Ca^2+^ addition) that was included previously. In subsection “Functional importance of Ca^2+^ binding to the Munc13-1 C_2_B domain” we describe that ‘the fluorescence increase reflecting lipid mixing was 28.9% of that observed 200 s after Ca^2+^ addition while content mixing was minimal at 300 s (fluorescence 3.3% of that observed 200 s after adding Ca^2+^)’. We also would like to emphasize that this difference is exacerbated by the fact that the maximal fluorescence associated with content mixing is expected to correspond to only one round of fusion, whereas additional rounds of fusion could contribute to the maximal fluorescence in the lipid mixing signal, which could easily correspond to more than 0.5 rounds of fusion if lipid mixing indeed arose from real fusion (which we do not believe). Hence, there is a dramatic difference in the amounts of lipid mixing and content mixing that occur before Ca^2+^ addition. Note also that the clear contrast between lipid and content mixing was also observed in the experiments without Syt1 or with Syt1 C_2_AB, in the presence of Munc18-1, NSF-αSNAP and C_1_C_2_BMUNC_2_C (Figure 4, blue curves). Moreover, the lipid and content mixing curves are much more similar to each other in experiments performed with T- and V-liposomes plus C_1_C_2_BMUNC_2_C (without Munc18-1 and NSF-αSNAP; see new Figure 5, blue curves), showing how content mixing occurs concomitantly with lipid mixing under these conditions but not when NSF-αSNAP and Munc18-1 are present. All of these observations and the fact that there is already extensive clustering before Ca^2+^ addition provide strong evidence that the drastic effect of Ca^2+^ in the presence of Munc18-1, NSF-αSNAP and C_1_C_2_BMUNC_2_C does not arise because of increased clustering but rather from a role of the Munc13-1 C_2_B domain in fusion and/or in the release of some inhibitory interaction. We have de-emphasized the proposal of a role for Munc13-1 in fusion throughout the Abstract and the Discussion, but we hope that the reviewers will agree that we can suggest these two possibilities based on the available data.

*3) Figure 6 suggests that C_1_C_2_BMUNC_2_C greatly enhances both content and lipid mixing in the absence of Ca^2+^ together with SNAREs, compared to just SNAREs alone. However, this result could also be a consequence of enhanced docking by C_1_C_2_BMUNC_2_C in the absence of Ca^2+^, rather than an effect on intrinsic spontaneous fusion itself. To assess this question, clustering experiments should be attempted with exactly the same conditions and constructs to assess if C_1_C_2_MUNC_2_C enhances vesicle clustering.*

We agree with the interpretation proposed by the reviewers, which is supported by the new clustering data that we provide (Figure 6 and several Figure 6 supplements).

*4) Figure 7 suggests that injection of Ca^2+^ does not induce any increase in lipid/content mixing in presence of SNAREs and full length Syt1 (orange curve, i.e., no Munc18, Munc13, αSNAP/NSF). This is at odds with previous reconstitution experiments that showed an increase of both properties with SNAREs and Syt1 alone upon Ca^2+^ injection (Malsam et al., EMBO J 2012; Lai et al., eLife 2014). An explanation is needed.*

Comparisons of our data with those obtained with single-vesicle assays such as those of Lai et al. 2014 are difficult because many differences have been observed previously between these assays and bulk assays (for instance in studies of complexins). There is no contradiction with the data of Malsam et al. as can be seen from Figure 1 of this paper. Note that small sudden jumps at the moment of Ca^2+^ addition are difficult to interpret because they may arise from temporarily high local Ca^2+^ concentrations when the Ca^2+^ is added, and that the curve after the jump continues as a natural extension of the curve before Ca^2+^ addition.

5) The effect of Munc13 in spontaneous (Ca^2+^ independent) and Ca^2+^-triggered fusion are very different when Syt1-C_2_AB fragment or reconstituted full-length Syt1 is used (compare Figure 5 and Figure 7), indicating that Syt1-C_2_AB behaves quite differently from full-length Syt1 in their reconstituted system. The use of reconstituted full-length Syt1 is more physiological. For the benefit of a clearer presentation of the results, all experiments with Syt1 C_2_AB could be eliminated from the paper.

See point E of the response to the Summary.

*6) Figure 6 and Figure 7 show the effect of C_1_C_2_BMUN on lipid mixing in the absence or presence of NSF/SNAP and Munc18-1, by the different effects of calcium in this context. The authors suggest that different mechanisms are at play, but there may be other possibilities. The calcium independent lipid mixing effects of C_1_C_2_BMUN are thought to reflect support of SNARE complex assembly by C_1_C_2_BMUN. In the presence of NSF/SNAP and Munc18-1 there are at least two explanations: (1) NSF/SNAP reverts the C_1_C_2_BMUN-mediated SNARE complex assembly in the absence of calcium, while in the presence of calcium more C_1_C_2_BMUN is recruited to the liposomes and can then override the antagonistic effect of NSF/SNAP. (2) The membrane recruited C_1_C_2_BMUN even protects assembled SNARE complexes from NSF/SNAP.*

We have considerably changed the interpretation of these experiments. We hope that the reviewers will agree with the current interpretation.

*7) Surprisingly, Syt1 does not appear to be essential to control either the speed or the calcium-sensitivity of the reaction when the longest forms of Munc13 are included in their assay (Figure 5). Thus, in this simplified system, the calcium-switch is encoded entirely within the C_2_domains of Munc13 at a calcium sensitivity somewhere below 100 μM. Establishing this limit with some precision would help ascertain how the activity of this system compares to the resting state in the synapse. The authors also speculate that Syt1-dependence may become apparent if they had faster time resolution in their assay. Given the central role of this calcium-sensor in synaptic biology, establishing whether it has any activity here is essential. However, as pointed out in (5) above, the results with the soluble Syt1 C_2_AB domain are rather different from the results with full-length Syt1. Thus, it might be prudent to eliminate all results with the Syt1 C_2_AB domain and focus on full-length reconstituted Syt1 throughout this work.*

It is true that the Ca^2+^ switch does come largely from the Munc13-1 C_2_B domain in these experiments (Figure 9). However, Syt1 C_2_AB does have an effect on content mixing in experiments with C_1_C_2_BMUN (Figure 4) and there also appears to be an effect of full-length Syt1 (compare Figure 4 and Figure 8), although this comparison is difficult to make because of the use of different proteoliposome populations. In the experiments with C_1_C_2_BMUNC_2_C, we cannot really expect much of an effect by Syt1 because the stimulation of fusion by Ca^2+^ is so fast in the time scale of our measurements. This is why we believe that techniques with faster time scales and that can distinguish docking from fusion, such as single vesicle assays, are more adequate to dissect the relative contributions of Munc13-1 and Syt1 to fusion.

We would like to clarify that the experiments of Figure 4—figure supplement 3 show that fusion is already maximized with 100 μM Ca^2+^ and 100 μM EGTA. Hence, the Ca^2+^ sensitivity is somewhere around 1 μM or below (not 100 μM or below). In the text we explain the difficulty of assessing this sensitivity more accurately because of the danger of removing the Zn2+ ions from the C_1_ domain. See also see point E of the response to the Summary with regard to the issue of removing data obtained with Syt1 C_2_AB.

*8) When shorter versions of Munc13 are tested where the C_2_C domain is missing, the authors now see Syt1-dependent effects on membrane fusion. In particular, they make the important point that lipid-mixing proceeds "efficiently" without Syt1, while contents-mixing is strongly enhanced by Syt1 C_2_AB. However, in Figure 4, it is not obvious that the effect of Syt1 is more pronounced for content vs lipid-mixing. Although the change in the curves for content-mixing appear more dramatic than for lipid-mixing, this seems to be due to the scale of the axes and not necessarily to the rates. The initial rate changes about 3 fold for contents mixing (comparing B and D). A similar increase in lipid mixing would be hard to see at their resolution, but looks plausible when comparing A and C. A similar burst in lipid-mixing appears to be present in Figure 5. Without meaningful resolution over this range, it is challenging to draw the conclusion on pg 12 "the finding that Syt1 C_2_AB selectively enhances content mixing but not lipid mixing provides strong evidence that Syt1 plays a direct role in membrane fusion…". The same point is echoed strongly in the Discussion. At a minimum, it would be helpful if the authors provided initial rates from each experiment, rather than endpoints. Otherwise, the discussion surrounding whether Syt1 is uniquely designed to support full-fusion should probably be more guarded. However, all these results were generated with the soluble Syt1 C_2_AB, so again, it might be better to eliminate all results with the Syt1 C_2_AB domain from this work.*

As pointed out in the response to point 4, we cannot draw conclusions from the small jumps observed upon Ca^2+^ addition. There are many ways of quantifying the lipid and content mixing assays, and we chose to quantify the data at 500 s because this represents a compromise that reflects in part the initial speed and in part the completion of the reaction at longer times. The quantification provided in Figure 4—figure supplement 1 strongly supports the conclusion that there is indeed an enhancement of content mixing by C_2_AB in the experiments performed with C_1_C_2_BMUN, while there is no clear enhancement of lipid mixing. We have reproduced these results in multiple additional experiments with other preparations, and the same conclusions are reached with the sulfhorhodamine assay (Figure 3—figure supplement 1; previously Figure 3). Nevertheless, we have drastically cut the discussion of these data because this is not a main point of this paper. We still conclude that C_2_ AB helps with membrane fusion in these experiments and cite other papers that have previously reached similar conclusions regarding the stimulation of fusion by Syt1.

*9) In the Introduction, the authors write "…the mechanism by which Munc13s play an additional role in vesicle docking… is unclear." The papers cited here do actually propose a mechanism for the role of vesicle docking by Munc13s, i.e. docking/membrane attachment, priming, and SNARE complex assembly may be manifestations of the same process, that is, that Munc13s mediate vesicle priming by promoting SNARE complex assembly (i.e., by opening the Munc18-syntaxin complex and catalyzing trans SNARE complex formation). The data provided in these papers are quite compelling. The three papers should be cited separately, including the corresponding conclusions, and then explain why the authors think that these papers are not helpful or believable in defining (part of) the mechanism by which Munc13 mediates vesicle docking. Somewhat in the same context, can one use the liposome-clustering activity of proteins and protein fragments as a basis to extrapolate towards corresponding roles of the respective proteins in synaptic vesicle docking in neurons? For example, in the case of Syt1, such a conclusion can probably not be drawn – loss of Syt1 does not really affect synaptic vesicle docking much.*

We have considerably revised the Introduction and the Discussion with regard to everything related to docking, as we now emphasize the role of Munc13s in docking much more than in the original manuscript. We have put our results in the context of different definitions of docking used in the literature and explain that our data are fully consistent with the interpretation of these three papers. In the text we indicate that these papers use a stringent definition of docking that then becomes equivalent to priming and SNARE complex assembly. However, we also propose that Munc13s are involved in upstream interactions with the vesicle and plasma membranes that help to bridge the membranes and facilitates the activity in promoting SNARE complex formation. With regard to the question about extrapolation, this is of course is a very important question and we believe that such extrapolations are more likely to be valid if they involve clear correlations between reconstitution and physiological data, and are consistent with other data in the field about the functions of the proteins under study (see point C of the response to the Summary).

[Editors' note: further revisions were requested prior to acceptance, as described below.]

1) The manuscript remains very long. Considering the revised conclusions, the authors are encouraged to take a fresh look and examine if all the data are really essential for the key conclusions.

We have considered this issue very carefully but we are reluctant to remove any of the data presented because, although some of the data may not be essential for the main conclusions, they still make important points that would be lost if the data are not presented here. We tried to reduce a little bit the length of the text while introducing changes to address the remaining concerns. While we agree that the manuscript is long, there are a lot of data that help building the overall story.

2) There are some supplementary figures that are actually quite important (e.g., Figure 4—figure supplement 4, and Figure 6—figure supplement 5). Please restrict supplementary figures to repeat experiments, raw data, and non-essential experiments.

The most critical panel of the previous Figure 4—figure supplement 4, panel C, has been moved to Figure 4 as panel E. The previous Figure 6—figure supplement 5 is now Figure 7 in the revised paper. As a consequence of this change, the previous Figure 7 is now Figure 8. To avoid increasing the overall number of figures, we have merged the previous Figure 8 and Figure 9 into a single new Figure 9. For consistency with this change, the figure supplements corresponding to the old Figure 8 and Figure 9 have been renumbered and are now Figure 9—figure supplement 1–Figure 9—figure supplement 3.

3) A general comment regarding all figures that compare lipid and content mixing results (Figure 4, Figure 5, Figure 7, Figure 8, Figure 9 and numerous supplemental figures): both types of experiments (lipid mixing and content mixing, respectively) are normalized differently: for lipid mixing 1% BOG is added and the resulting fluorescence intensity value is used for normalization; whereas for content mixing, it is the maximum value for each group of conditions in the particular panel. So, 100% content mixing could actually correspond to substantially fewer vesicles undergoing content mixing than vesicles undergoing lipid mixing. A suggestion would be to change all figure panel titles from "lipid mixing" to "relative lipid mixing" and "content mixing" to "relative content mixing", and to change all the y-axis labels from "Fluorescence (% of max)" to "Fluorescence (arbitrary units)". Another comment: ensemble experiments cannot determine what fraction of vesicles that undergo lipid mixing also undergo content mixing.

The terms ‘lipid mixing’ and ‘content mixing’ in the figures are just meant to be used as titles for easy identification of the data by the reader. We believe that adding the term relative may be more confusing than clarifying. The key issue that we agree needs to be made clearer is why the fluorescence signal reflecting lipid mixing is much smaller than that reflecting content mixing when both are expressed as percentages of the maximum fluorescence. We have clarified this issue in the text with the following paragraph:

‘Note that in these assays much of the Cy5 fluorescence increase caused by FRET from PhycoE (reflecting content mixing) should occur during the first round of fusion and that no further substantial increases are thus expected in subsequent rounds of fusion or upon detergent addition. Correspondingly, the maximum Cy5 fluorescence observed in our most efficient fusion reactions was similar to that observed upon detergent addition (e.g. Figure 3, red curve; see Materials and methods). In contrast, the lipid mixing signal expressed as percentage of maximum Marina Blue fluorescence is much smaller in the same reactions (e.g. Figure 3, red curve) because fluorescence de-quenching is expected to continue in successive rounds of fusion and to undergo a further, large increase upon detergent addition due to additional probe dilution.’

4) Figure 4—figure supplement 4. We appreciate that the authors have performed DLS measurements to address the concern that the large increase in content mixing upon calcium addition could be due to a calcium-dependent docking/clustering effect. However the presence of large vesicles might dominate in the intensity autocorrelation functions shown. Percent Intensity plots (similar to those shown in Figure 2 and Figure 6) should also be provided for Figure 4—figure supplement 4.

We include percent intensity plots for the most critical DLS data in panels C-F of the revised Figure 4—figure supplement 4.

5) Figure 4—figure supplement 4 was only done for the soluble Syt1 C_2_AB fragment. If at all possible, please also perform the DLS experiments also with full-length Syt1. The reason is that there is much less lipid mixing in these experiments prior to calcium addition than for the experiments with the soluble Syt1 C_2_AB fragment (compare Figure 7 and Figure 8).

The suggested experiments with full-length Syt1 were already described in the previous manuscript and shown in the old Figure 9—figure supplement 2, which is now Figure 9—figure supplement 3 in the revised manuscript (because of the changes summarized in point 2).

*6) Figure 9: It is perhaps possible that there could be compensating factors in neurons that do not show a large phenotype for the DN mutant of Munc13. Nevertheless, if the author's model of calcium-independent bridging of T and V-vesicles by C_1_C_2_BMUNC_2_C is correct (Figure 6—supplement figure 5), why would calcium binding to the C_2_B domain of Munc13 have such a large effect? Please discuss.*

We briefly discussed in the previous manuscript that

‘…Our data suggest that the primed state is metastable and requires Ca^2+^ binding to the Munc13-1 C_2_B domain for efficient content mixing (Figure 9) either because the Ca^2+^-bound Munc13-1 C_2_B domain contributes directly to facilitate membrane fusion or because Ca^2+^ binding releases an inhibitory interaction existing in the primed state. The finding that a mutation in a Ca^2+^-binding loop of the Munc13-2 C_2_B domain increases release probability (Shin et al., 2010) is consistent with both possibilities and supports the notion that Munc13s form intrinsic part of the primed state of synaptic vesicles.’

We have left this same paragraph in the revised manuscript without additional discussion to avoid further lengthening of the text.

7) Figure 6—figure supplement 5: While the preference for the C_1_C_2_B domain for the T-vesicle membrane can be explained by the presence of DAG and PIP2 in that membrane, what is the mechanism that C_2_C would only interact with the V-vesicle membrane?

The model proposed in the new Figure 7, which was previously Figure 6—figure supplement 5, assumes that the C_1_ and C_2_ B domains bind to the plasma membrane and such binding places Munc13 in an orientation that favors binding of the C_2_C domain in trans to another apposed membrane (the synaptic vesicle membrane in this case). The model is discussed in pages 23-24, where we make it clear that the mechanism of liposome clustering by these large Munc13 fragments remains to be established, as there are multiple sites in these fragments that could potentially interact with membranes. Hence, extensive studies will be needed to define this mechanism clearly.

8) Figure 2 and Figure 6: why does C_1_C_2_BMUN cluster V-vesicles, whereas C_1_C_2_BMUNC_2_C does not? This question is also related to the Discussion section, paragraph five. Please provide an explanation why the C_2_C domain would favor trans-interactions with V-vesicles.

In the previous manuscript, we had provided an explanation for the DLS data but not a model figure because there are multiple potential explanations and we preferred not to favor too much one over another. In response to this concern, we have added a new Figure 7—figure supplement 1 in the revised manuscript that describes plausible models to explain the DLS data. In the figure legend we make it clear that there are other potential models, as we do not want to mislead the reader into thinking that the proposed models are well established.

9) Subsection “Simultaneous evaluation of lipid and content mixing”, paragraph two": "A problem…". Actually, synaptic vesicles are acidic, so the property of sulforhodamine to produce an acidic interior of the proteoliposomes is not necessarily a problem. Moreover, the lack of leakiness under such conditions is desirable. In any case, it is good that the authors confirmed the content mixing results with the different content mixing assay by Zucchi and Zick. Thus, there is really no "problem" with the sulforhodamine content mixing assay.

We agree that the acid interior of the liposomes is not necessarily a problem, but the pH in the concentrated sulforhodamine solution that becomes encapsulated into the liposomes is very acidic (pH 2). We devoted extensive efforts to do fusion assays with sulforhodamine in both vesicle populations and were never able to observe efficient lipid mixing. In any case, we have replaced the word ‘problem’ with ‘potential concern’ in that sentence to address this issue.

10) Figure 1: the effect of the different lipid compositions (PIP2 and DAG) appears to be much more pronounced in the presence of NSF/SNAP/Munc18, although C_1_C_2_BMUN is present in both experiments.

Yes, this is an important point that we emphasized in the previous manuscript and still emphasize in the revised manuscript in the sentence (Results section):

‘These data show that the effect of Munc13-1 C_1_C_2_BMUN alone on lipid mixing arises from a property that is largely independent of Ca^2+^, DAG and PIP_2_, whereas the lipid mixing observed in the more complete reconstitutions including C_1_C_2_ BMUN, Munc18-1 and NSF-αSNAP is stimulated by Ca^2+^, DAG and PIP_2_, thus exhibiting properties that are more similar to those of neurotransmitter release.’

11) Discussion, paragraph seven: in addition to Lee and Kyoung, please also cite the improvements of this assay by Diao et al. (eLife 2012); Lai et al. (eLife 2014). Another comment: the single vesicle content mixing assay by Kyoung et al. (2011); Lai et al. (2014) allows discrimination between effects that are due to docking vs. due to fusion probabilities in addition to the improved time resolution offered by such single vesicle assays.

We have included the additional requested references.

[Editors' note: further revisions were requested prior to acceptance, as described below.]

We thank the authors for their thoughtful considerations of the concerns raised previously. However, the Reviewing Editor has two remaining concerns:

1) Related to Point 9, the authors write:

"A potential concern with the use of sulforhodamine B fluorescence de-quenching to monitor content mixing in these bulk reconstitution experiments is that de-quenching can also arise from liposome leakiness. We attempted to perform experiments where both liposome populations are loaded with sulforhodamine B to assess how much of the de-quenching arises from leakiness (Yu et al., 2013). However, we were unable to observe lipid mixing in these control experiments. Since the solutions of high sulforhodamine B concentrations trapped in the liposomes are highly acidic (pH about 2), which may underlie the lack of lipid mixing, we attempted to increase the pH of these solutions to 5.5 or 7.4 before trapping them into the liposomes, but at both pH values the liposomes were strongly leaky."

Their observation of lack of lipid mixing when both vesicle populations are filled with sulforhodamine B is strange, considering that the authors observed efficient lipid mixing when only one class of vesicles is filled (Figure 3—figure supplement 1). Moreover, Kyoung et al., 2011 and Diao et al. 2012 observed both lipid and content mixing using their sulforhodamine B content mixing method, with only an extremely small fraction of leakage (0.01% of the docked vesicles). Could the problems be related to differences in reconstitution methods, rather than an issue with the sulforhodamine method per se?

This Reviewing Editor also would like to point out that the pH of the interior of the synaptobrevin/synaptotagmin vesicles ("donor" vesicles) is likely more neutral rather than pH 2 when using the published reconstitution protocol (Kyoung et al. Nat Protoc 8(1):1-16, 2013): the initial protein/lipid/detergent solution uses a detergent concentration that is at the CMC (OG ~ 30 mM), so vesicles do not yet form at this stage. This solution is then injected on a Cl^-^4B column that has been pre-equilibrated with vesicle buffer (pH 7.4, without sulforhodamine B), followed by elution with vesicle buffer (pH 7.4, again without sulforhodamine B) (Kyoung et al. Nat Protoc 8(1):1-16, 2013). Vesicles are forming on the column in a time course of several minutes as the detergent concentration decreases. One would expect that the pH of the interior of the column rapidly reaches 7.4.

In any case, as the authors point out, the alternative PhycoE-Biotin/Cy5-Streptavidin content mixing assay produces similar results (Figure 3—figure supplement 3). However, as written, the second paragraph of subsection “Simultaneous evaluation of lipid and content mixing” may come across to the casual reader as if there is something wrong with the sulforhodamine B content mixing method. Since this is a rather technical issue that distracts from the main messages of this paper, and that may be related to differences in reconstitution methods, the authors may wish to consider deleting this entire paragraph. In the future, perhaps the authors may wish to investigate a variety of content mixing assay jointly with other groups who have developed these assays for a more technical publication.

We would like to clarify that in the first revision we insisted in keeping these data because the first round of review questioned the conclusion that synaptotagmin C_2_AB was required for efficient content mixing but not lipid mixing in the reconstitutions including C_1_C_2_BMUN, and the sulforhodamine assays provided further support for this conclusion. In the second revision we did not remove the data obtained with the sulforhodamine method because the argument made in the second round of review (stating that there is no problem) was not compelling; we did have a big problem with the method when we included sulforhodamine on both vesicles. With regard to the concerns raised in this third round of review, it is unclear whether the buffer can handle the strongly acidic pH caused by the very high concentrations of sulforhodamine that are being used. However, we agree that the problems that we described for the sulforhodamine method could represent a distraction. Hence, we have removed these data in the third revised manuscript.

2) Point 11, related to the sentence starting "Approaches that allow faster measurements [e.g. single vesicle assays (Lee et al., 2010; Kyoung et al., 2011); will be required to test this prediction. " It is not just the faster time resolution that is important for future studies. The single vesicle-vesicle lipid/content mixing assay by Kyoung et al. 2011, Diao et al., 2012 et al. (but not the assay by Lee et al., 2010) can distinguish between effects due to docking, lipid mixing and content mixing on a 100 msec time scale for each individual vesicle pair. In addition to the improved time resolution, the discrimination between effects due to docking and/or fusion will be important for future studies. The authors may wish to consider making the discussion of the relevant publications clearer.

After the aforementioned sentence, we have now added the following: ‘An additional advantage of these assays is that they allow distinction of the docking event from lipid and content mixing.’